EMBO
Molecular Medicine

# Histidine-rich glycoprotein modulates neutrophils and thrombolysis-associated hemorrhagic transformation

Wei Jiang [1,6], Yuexin Zhao [1,6], Rongrong Liu [1,6], Bohao Zhang[2], Yuhan Xie[3], Bin Gao[4], Kaibin Shi[4], Ming Zou[1], Dongmei Jia[1], Jiayue Ding[1], Xiaowei Hu[1], Yanli Duan[1], Ranran Han[1], DeRen Huang[1], Luc Van Kaer [5] & Fu-Dong Shi [1,4 ✉]

## Abstract

Intravenous thrombolysis using recombinant tissue plasminogen activator (tPA) remains the primary treatment for patients with acute ischemic stroke (AIS). However, the mechanism of tPA-related hemorrhagic transformation (HT) remains poorly understood. Elevation of histidine-rich glycoprotein (HRG) expression was detected by nano-liquid chromatography tandem mass spectrometry at 1 h following tPA infusion as compared to baseline prior to tPA infusion (discovery cohort, $n = 10$), which was subsequently confirmed in a validation cohort ($n = 157$) by ELISA. Surprisingly, no elevation of HRG was detected in individuals who subsequently developed HT. During in vitro experiments, HRG reduced neutrophil NETosis, inflammatory cytokine production, and migration across the blood–brain barrier induced by tPA. In a photothrombotic murine AIS model, HRG administration ameliorated HT with delayed thrombolysis, by inhibiting neutrophil immune infiltration and downregulating pro-inflammatory signaling pathways. Neutrophil depletion or NETosis inhibition also alleviated HT, whereas HRG siRNA treatment exacerbated HT. In conclusion, fluctuations in HRG levels may reflect tPA therapy and its associated HT. The inhibitory effect of HRG on neutrophils may counteract tPA-induced immune abnormalities and HT in patients with AIS.

Keywords tPA; Histidine-rich Glycoprotein; Hemorrhagic Transformation; Ischemic Stroke; Neutrophil
Subject Categories Immunology; Neuroscience

## Introduction

Stroke is the second leading cause of death worldwide. Acute ischemic stroke (AIS) constitutes ~70% of all strokes and is characterized by high disability and mortality rates and, consequently, a substantial economic burden (2021, Wang et al, 2017). Intravenous thrombolysis is the primary approach for the treatment of AIS, and recombinant tissue plasminogen activator (tPA), which is commercially available as alteplase, is currently the only FDA-approved thrombolytic agent (Powers et al, 2019). Although new treatment strategies aimed at improving recanalization are under development, tPA administration within 4.5 h of symptom onset remains the primary approach for early reperfusion therapy (Powers et al, 2019; Tsivgoulis et al, 2023). Despite its efficacy, the therapeutic time window of tPA is limited to 4.5 h, after which tPA administration causes heightened frequency of hemorrhagic transformation (HT). Delayed tPA treatment is associated not only with poor functional recovery but also with high rates of HT, so that only a small portion of AIS patients are eligible for and benefit from early thrombolysis (Hacke et al, 1995). Currently, less than 5% of patients with AIS benefit from early thrombolysis in clinical practice (Thiebaut et al, 2018).

The pathophysiology of tPA-related HT remains to be defined, which might involve ischemic injury, consumptive coagulopathy, reperfusion injury, and disruption of the blood–brain barrier (BBB) (Yaghi et al, 2014). Our previous study revealed that immune cell mobilization contributes to the exacerbation of HT and edema following thrombolysis (Shi et al, 2021). Neutrophil mobilization can be observed as early as 1 h after completion of tPA infusion, likely through the tPA receptors expressed by these cells (Shi et al, 2021).

The immune and coagulation systems interact and respond to each other through several linked components (Delvaeye and Conway, 2009). Administration of tPA induces a coagulative/fibrinolytic cascade with potential long-lasting effects on the immune system rather than transient, direct tPA-mediated immune cell interactions, which in turn affects the thrombolytic outcome in

[1]Department of Neurology, Tianjin Neurological Institute, Tianjin Medical University General Hospital, Tianjin 300052, China. [2]Department of Radiology, The Third Affiliated Hospital of Zhengzhou University, Zhengzhou 450052 Henan, China. [3]Department of Neurology, Tianjin NanKai Hospital, Tianjin 300102, China. [4]Center for Neurological Diseases, China National Clinical Research Center for Neurological Diseases, Beijing Tiantan Hospital, Capital Medical University, Beijing 100070, China. [5]Department of Pathology, Microbiology and Immunology, Vanderbilt University School of Medicine, Nashville, TN 37232, USA. [6]These authors contributed equally: Wei Jiang, Yuexin Zhao, Rongrong Liu. ✉E-mail: fshi@tmu.edu.cn

a feedback loop. The half-life of tPA is 4 min, which indicates >80% clearance within 10 min (Gillis et al, 1995; Matosevic et al, 2013). Despite its short half-life in the bloodstream, effects of tPA on the coagulation system persist for more than 24 h after infusion (Matosevic et al, 2013). Thus, we speculated that the coagulative/fibrinolytic cascade induced by tPA administration might have sustaining effects on immune modulation. However, the role of fibrinolysis/coagulation-related products induced by tPA in immune modulation and HT remains unknown.

In this study, high-throughput proteomic technology was employed for protein screening to determine tPA-induced protein variations in the plasma of patients with AIS, which revealed induction of histidine-rich glycoprotein (HRG). Although HRG can play multifaceted functions in coagulation/fibrinolysis processes and infectious diseases (MacQuarrie et al, 2011; Malik et al, 2023; Nishibori et al, 2018; Shigekiyo et al, 1998; Vu et al, 2015), its effect during thrombolysis is unclear. We sought to explore the role of HRG in the innate immune system and HT following intravenous thrombolysis with tPA.

## Results

### Proteomic analysis of differentially expressed proteins (DEPs) reveals a significant elevation of HRG after tPA treatment

Ten patients (4 women, 6 men; average age of 64.4 years) with AIS who underwent tPA treatment were included in the proteomic profiling analysis. All patients received tPA treatment within the thrombolytic time window of 4.5 h. Subsequently, two patients developed HT with no symptomatic exacerbation. The basic clinical data of these patients are presented in Table EV1. Blood samples were collected from each patient prior to the initiation of tPA infusion and at 1 h after completion of tPA treatment. We performed high-throughput MS-based proteomics using a DIA strategy for plasma proteome profiling. A total of 22,274 peptides and 4602 proteins were identified and quantified in the samples. Since samples were collected before and after treatment from every patient, paired comparisons were made based on proteomic quantification. The DEPs were identified and defined as a fold change >1.5 and Q-value < 0.05. A total of 119 DEPs were identified, including 84 upregulated and 35 downregulated proteins, as shown in Fig. 1A. DEPs were analyzed using KEGG classification and were found to participate mainly in the immune system (Fig. 1B). KEGG pathway analysis further showed that DEPs were significantly enriched in pathogenic infection, gap junction, platelet activation, complement and coagulation cascade, and TGF-β signaling (Fig. 1C). GO enrichment analysis indicated that the DEPs were significantly enriched in proteins involved in certain biological processes, including the mitotic cell cycle, microtubule-based processes, negative regulation of endothelial cell chemotaxis, platelet activation, and fibrinolysis (Fig. 1D). Thus, KEGG and GO enrichment analyses of DEPs indicated that tPA induces coagulative, fibrinolytic, immune, and vascular/endothelial responses following tPA infusion. The top 20 most significantly upregulated proteins were identified based on −log10-transformed Q-values (Fig. 1E). The fold-change rankings of the proteins are presented in Fig. 1F. HRG, plasminogen (PLG), Fibrinogen alpha chain (FGA) and CPB2 were among the proteins that were increased drastically. PLG and FGA levels have been widely reported to increase after ischemic stroke and are produced after thrombolysis (Lee et al, 2020; Tsurupa and Medved, 2001). CPB2 participates in the downregulation of fibrinolysis by removing the C-terminal lysine residues from fibrin, which is partially degraded by plasmin (Mao et al, 1999).

Next, we performed PPI network analysis of the above identified DEPs using the STRING database. After removing proteins with no identified symbol name, 49 nodes and 196 edges were involved in the PPI network (Fig. EV1A1). The top 20 hub DEPs identified using the CytoHubba plugin included ACTB, FN1, CFL1, ACTA2, RHOA, TAGLN, YWHAZ, THBS1, HSP90AB1, TUBA1C, TUBA1B, TUBB, KNG1, ITIH4, TAGLN, MYL6, FGA, FGG, HRG, TPM4, and F5 (Fig. EV1B). The MCODE plugin distinguished two cluster networks, one of which was mainly related to fibrinolysis and inflammatory responses. Seven of the top 20 hub genes, namely HRG, FGG, FGA, F5, KNG1, THBS1, and ITIH4 were included in the cluster (Fig. EV1C,D). HRG appeared as a central node in the interaction network connecting proteins related to fibrinolytic and inflammatory processes. Therefore, we sought to elucidate the role of HRG increments following tPA treatment in patients with AIS.

### HRG production following tPA treatment is not elevated in individuals with HT

Elevated HRG levels in plasma were detected at 1 h after thrombolysis by LC-MS in our discovery cohort. We sought confirmation in another cohort of patients with AIS. In addition, whether the increased levels of HRG were induced by tPA treatment or by the progression of AIS remained unclear. To further clarify this issue, patients with AIS, with or without tPA treatment, and healthy controls (HC) were included in this study. Paired blood samples were collected prior to the initiation and at 1 h after the completion of tPA infusion. Patients who declined or were not eligible for thrombolysis were recruited into a (AIS w/o tPA) group, and blood samples from these patients were drawn at 3.5 ± 0.5 h and 6.5 ± 0.5 h post stroke symptom onset. The clinical features of the patients are presented in Table 1. Detailed information on these individuals from the validation cohort is presented in Tables EV2–5. Again, HRG levels were significantly elevated in the tPA-treated group at 1 h post-tPA infusion; however, no significant changes in HRG levels were observed in the AIS group who received no tPA treatment, and instead received antiplatelet treatment (Fig. 2A). This implies that the increase in plasma HRG can be specifically attributed to tPA treatment, rather than to the progression of stroke. To further investigate the potential role of HRG elevation following thrombolysis, we examined the relationship between the increase in HRG levels at 1 h post-tPA infusion and subsequent occurrence of HT or post-stroke infection. We found a correlation between lack of increase in HRG levels at 1 h following tPA infusion and HT within 24 h of tPA treatment (Fig. 2B). Lower levels of HRG at 1 h after tPA treatment predict increased risks of HT (Fig. 2C). No differences in HRG levels were observed between patients who did or did not develop post-stroke infections (Fig. 2D). Elevation of plasma HRG level immediately following intravenous tPA treatment was further observed in a murine model of AIS (Appendix Fig. S1). Together with our findings in LS-MS analyses, these results suggest that HRG production following tPA infusion may exert a protective role by reducing the risk of tPA-associated HT.

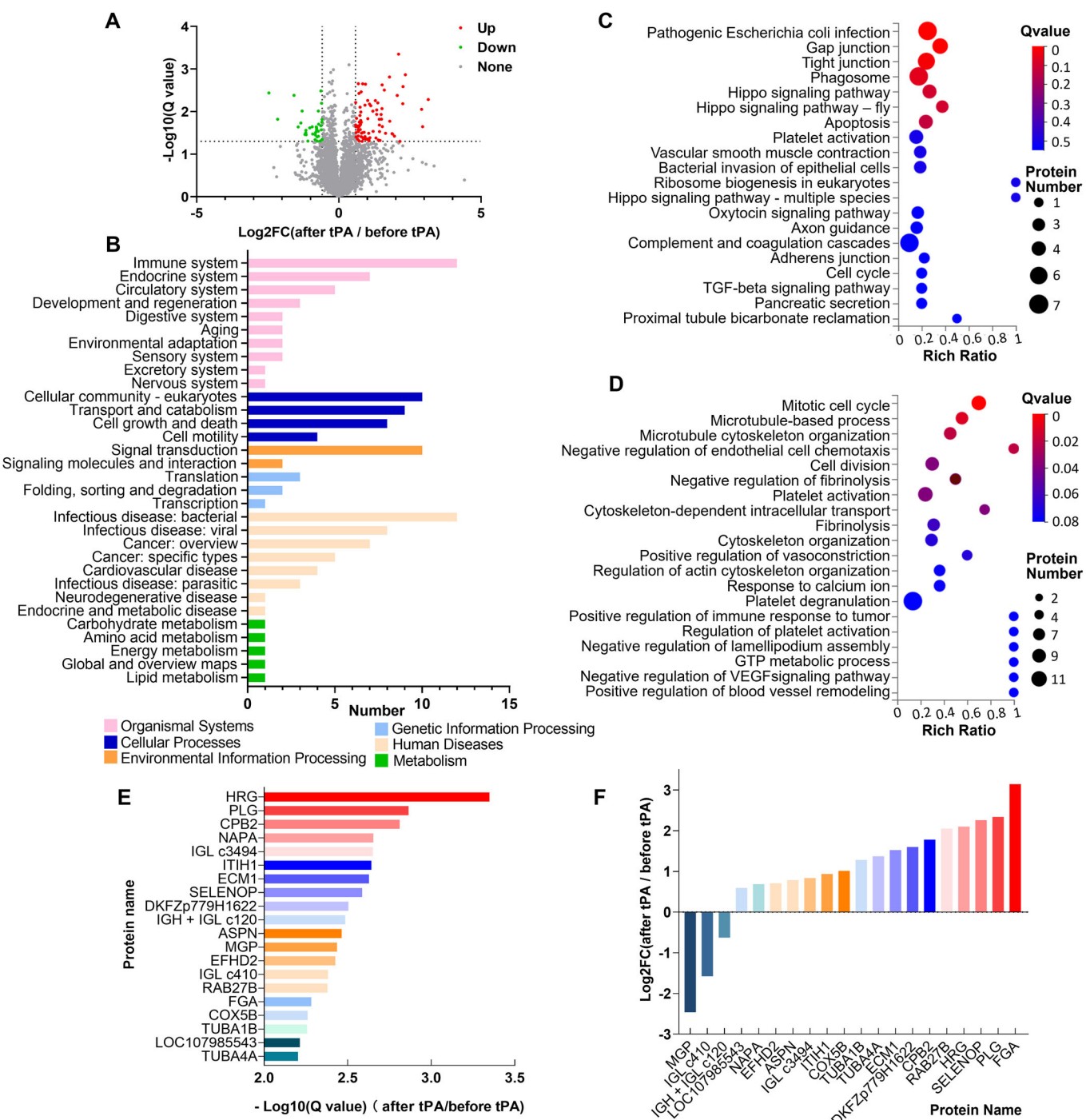

**Figure 1. Identification of differentially expressed proteins in plasma of patients following recombinant tissue plasminogen activator (tPA) treatment.**

(A) Protein profiles from the paired plasma samples at baseline and 1 h after tPA treatment were detected by the data-independent acquisition (DIA) mode ($n = 10$). A total of 119 DEPs were identified (fold change >1.5 and Q-value < 0.05), including 84 significantly upregulated proteins and 35 significantly downregulated proteins. Volcano plot showing the significance vs. fold change before and after tPA treatment. The X-axis represents protein difference (log2-transformed fold changes), and the Y-axis represents the −log10-transformed Q-values (adjusted p-values). Red dots indicate significantly upregulated proteins, green dots indicate significantly downregulated proteins, and gray dots symbolize proteins with no significant change. (B) The DEPs were analyzed by KEGG classification in individuals after vs. before tPA treatment. (C) Bubble plot of KEGG pathway analysis of DEPs. Rich ratio represents the ratio of the number of target proteins vs. all proteins included in each pathway. Dot size represents the number of enriched proteins. The color of the dot represents the size of the Q-value. (D) Bubble plot of GO enrichment analysis of DEPs. Dot size represents the number of enriched proteins. The color of the dot represents the size of the Q-value. (E) The top 20 most significantly increased DEPs are shown based on the −log10-transformed Q-values. (F) The fold change of the DEPs in individuals after vs. before tPA treatment, ranked according to the log2-transformed fold change.

## HRG maintains neutrophil viability after tPA stimulation in vitro

To investigate the role and mechanism of HRG in immune modulation, we focused on peripheral blood cells and cytokines in the presence of tPA. Mononuclear cells and neutrophils were isolated from human peripheral blood and treated with various concentrations of tPA. After an 18-h incubation period, the cells were stained with calcein-AM and Hoechst 33342, and cell viability was examined and quantified by fluorescence microscopy (Fig. 3A). The results demonstrated that the rate of neutrophil viability decreased following tPA stimulation, with a higher concentration of tPA corresponding to a greater decrease in cell viability (Fig. 3B). In addition, the overall neutrophil viability decreased gradually with prolonged incubation time (Appendix Fig. S2). However, the viability rate of the mononuclear cells did not change in response to tPA treatment (Appendix Fig. S3). Different concentrations of HRG were added to the neutrophil cultures treated with 10 μg/ml tPA (Fig. 3C,D). Interestingly, co-treatment with HRG resulted in reversal of tPA-related neutrophil viability in a dose-dependent pattern, with higher concentrations of HRG exhibiting a more pronounced improvement in cell viability (Fig. 3D,E). In addition, the decreased neutrophil viability during prolonged incubation times was also reversed by HRG treatment (Appendix Fig. S2). These results suggest that in AIS, tPA may have a pronounced impact on neutrophil function and survival as compared to mononuclear cells, whereas HRG may have a protective effect on maintaining neutrophil survival in the presence of tPA.

## HRG reduces tPA-related NETosis through CLEC1B and inhibiting excessive inflammatory responses in neutrophils

Neutrophil cell death is primarily induced by exogenous stimuli that result in inflammatory exhaustion, referred to as NETosis. To further investigate the effects of HRG and tPA on neutrophils, cells were stimulated with tPA for 2.5 h in vitro, and followed by immunofluorescence co-staining with anti-H3Cit, anti-MPO, and DAPI. NET formation by neutrophils was observed and quantified using fluorescence microscopy, with LPS added as a positive control (Fig. 4A). Our findings revealed a significant increase in NETosis following tPA stimulation. Addition of HRG resulted in a sharp decrease in NETosis (Fig. 4A–D). Supernatants from the culture system were collected to measure the concentrations of inflammatory cytokines. HRG inhibited the production of TNF-α, IL-8, and ROS levels induced by tPA treatment (Fig. 4E–G). These results demonstrated that tPA treatment activates neutrophil inflammatory responses, whereas HRG administration reduces the excessive activation of neutrophils and NETosis after tPA treatment and inhibits the release of ROS and inflammatory cytokines.

HRG has been reported to bind to various components and receptors including the C-type lectin receptors CLEC1A (Gao et al, 2020; Takahashi et al, 2021) and CLEC1B (Nishimura et al, 2019). Thus, to identify the specific receptors on neutrophils engaged by HRG, blocking antibodies against CLEC1A and CLEC1B were added into the tPA-treated neutrophil cultures, with HRG added subsequently (Fig. EV2). The decreased mortality rate induced by HRG was reversed in the anti-CLEC1B Ab but not the anti-CLEC1A Ab treatment group, indicating that CLEC1B might be involved in mediating the protective effects of HRG towards tPA-

**Table 1. Demographic characteristics of AIS patients in the validation cohort.**

|  | Healthy control (n = 53) | Stroke patients without tPA treatment (n = 42) | Stroke patients with tPA treatment (n = 62) | P |
|---|---|---|---|---|
| Age (years) | 66.0 ± 9.4 | 68.3 ± 11.7 | 65.5 ± 9.6 | ns |
| Sex, female (%) | 30.2 | 33.3 | 27.4 | ns |
| Risk factors |  |  |  |  |
| Cardiovascular disease (%) | 22.6 | 14.3 | 19.4 | ns |
| Hepertension (%) | 66.0 | 59.5 | 59.7 | ns |
| Diabetes (%) | 35.8 | 33.3 | 27.4 | ns |
| Atrial fibrillation (%) | 11.3 | 16.7 | 17.7 | ns |
| Smoking (%) | 47.2 | 50.0 | 53.2 | ns |
| Drinking (%) | 47.2 | 52.4 | 43.5 | ns |
| NIHSS score | / | 5.5 ± 4.8 | 7.0 ± 5.7 | ns |
| Time of onset (h) | / | 3.0 ± 0.8 | 3.2 ± 0.8 | ns |
| Hemorrhagic transformation |  |  |  |  |
| No (%) | / | 97.6 | 88.7 | ns |
| Yes (%) | / | 2.4 | 11.3 | ns |

stimulated neutrophils (Fig. EV2B). Furthermore, NET formation was evaluated in the anti-CLEC1A and anti-CLEC1B Ab treatment groups (Fig. EV2C). We found that the decreased NET formation in response to HRG treatment was reversed in the anti-CLEC1B Ab treatment group, but not the anti-CLEC1A Ab treatment group (Fig. EV2D–F). Therefore, the above results indicated that CLEC1B might play an essential role in mediating the protective effect of HRG towards neutrophil NETosis.

## HRG inhibits neutrophil F-actin polymerization and migration across the BBB in vitro

Neutrophil infiltration into the central nervous system is crucial for the progression of AIS following thrombolysis. Thus, we investigated the effects of tPA and HRG on neutrophil migration. Neutrophils isolated from human peripheral blood were co-cultured with tPA for 2.5 h, and F-actin polymerization was determined by phalloidin staining under a fluorescence microscope (Fig. 5A). A significant increase in F-actin polymerization was observed with tPA treatment. However, polymerization was significantly reversed in the presence of HRG (Fig. 5B). These findings suggest that tPA induces cytoskeletal deformation in neutrophils, rendering them more susceptible to migration, and that this deformation can be inhibited by HRG. Furthermore, we simulated the BBB in vitro by coating Transwell chambers with hCMEC/D3 brain endothelia cells and subjecting them to oxygen-glucose deprivation for 4 h to mimic ischemic hypoxia in ischemic stroke. Neutrophils pretreated under different experimental conditions (control, tPA, tPA + 0.1 μM HRG, tPA + 0.5 μM HRG, or tPA + 1 μM HRG) for 2.5 h were added into the upper chamber. The percentage of neutrophil migration across various time points was assessed (Fig. 5C). HRG significantly inhibited the percentage

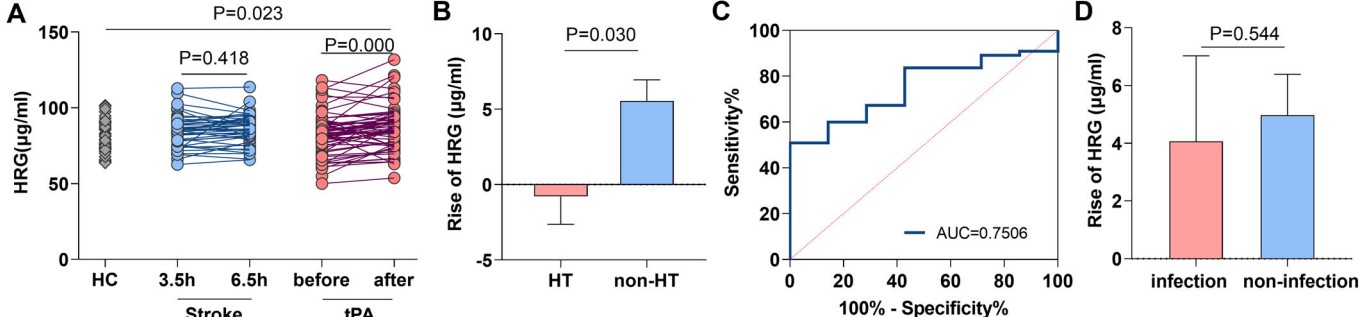

**Figure 2. Validation of increased levels of histidine-rich glycoprotein (HRG) in plasma following tPA and its relevance to clinical prognosis.**

(A) 53 healthy volunteers were included as controls. In the tPA untreated group, plasma samples were collected at 3.5 h and 6.5 h after stroke onset for HRG determination (n = 42). In the tPA-treated group, HRG levels were detected at baseline and 1 h after tPA treatment for paired comparison (n = 62). (B) Patients with tPA treatment were further divided into two groups based on the complication of hemorrhagic transformation (HT): HT (n = 7), non-HT (n = 55). The levels of HRG elevation after thrombolysis were compared between the two groups. (C) The Receiver operating curve (ROC) analysis of HRG for prediction of HT in patients following tPA treatment was shown. The area under the curve (AUC) in ROC curve analysis was 0.7506. (D) Patients with tPA treatment were further divided into two groups based on the complication of post-stroke infection: infection (n = 9), and non-infection (n = 53). The levels of HRG elevation after thrombolysis were compared between the two groups. Data information: Data are presented as mean ± SEM; two-tailed paired Student's t test (3.5 h and 6.5 h; before and after) and Two Independent-Samples t test (HC and Stroke; HC and tPA; Stroke and tPA) are used in (A). Two-tailed Mann–Whitney U-test is used in (B, D).

of migrated neutrophils at 2 h, 4 h, and 8 h after loading (Fig. 5D–F). In addition, the speed of neutrophil migration (percentage of migrated cells within a specific time window) was highest within the first 2 h in the transwell assay, and decreased gradually at 2–4 h and even more profoundly at 4–8 h. The inhibitory effect of HRG on the neutrophil migration rate was obvious within the first 2 h of culture, a time frame likely optimal for evaluating the migration of viable and motile neutrophils (Fig. 5G,H). These results indicated that HRG reduces the migratory capacity of neutrophils to cross the BBB in vitro.

## HRG reduces tPA-related HT by inhibiting the infiltration of neutrophils in a murine acute ischemic stroke model

To further investigate the role of HRG in vivo, we designed an experiment using a murine AIS model as in Fig. 6A. Briefly, a model of photothrombotic middle cerebral artery occlusion (MCAO) was established in C57BL/6 mice, with tPA administered separately at 1 h and 5 h, corresponding to early and delayed thrombolysis. The HRG levels in MCAO, MCAO + tPA(1 h) and MCAO + tPA(5 h) were determined, which showed a significant increase at 1 h after tPA treatment that were maintained at high levels until 5 h post-stroke onset followed by a gradual decrease to normal levels at 24 h. Delayed initiation of HRG induction was found in the MCAO + tPA(5 h) group, which reached high HRG levels at 2 h after tPA treatment (Fig. EV2). Thus, HRG supplementation was administered in combination with delayed thrombolysis to evaluate its effects in vivo. Twenty-four hours after stroke onset, the Bederson neurological score, rotarod test, and corner test were performed to evaluate sensorimotor functions of the mice, which revealed that early tPA treatment (at 1 h) effectively improved the neurological function of mice 24 h after stroke. However, no benefits were observed in the mice treated with tPA at 5 h after the onset of stroke. Interestingly, supplementation with HRG in combination with delayed tPA treatment significantly improved neurological function (Fig. 6B–D). Mouse brains were harvested 24 h post-stroke after intracardial perfusion with PBS.

Compared with the early tPA treatment group, the delayed tPA treatment group displayed worse HT. A marked reduction in hemorrhagic volume was observed with the addition of HRG (Fig. 6E). The hemoglobin concentration in the HRG treatment group was much lower than that in the group without HRG treatment at 5 h post stroke onset (Fig. 6F). Finally, the extent of neutrophil infiltration in the lesions of each group was assessed using immunofluorescence staining, which revealed a significant increase in neutrophil infiltration after tPA treatment that was reversed by the addition of HRG (Fig. 6G,H). Therefore, HRG may reduce tPA-associated HT by inhibiting neutrophil infiltration.

We used siRNA for depletion of endogenous HRG and investigated its effect on HT. HRG siRNA treatment efficiently reduced HRG mRNA expression in AML12 hepatocyte cells in vitro (Appendix Fig. S4A). In addition, HRG siRNA treatment reduced the expression of HRG mRNA in liver (Appendix Fig. S4B) and rapidly reduced plasma HRG levels in vivo (Appendix Fig. S4C). Next, we treated mice with HRG siRNA, induced MCAO, and performed delayed tPA treatment. We found that neutrophil infiltration, HT and neurological deficits were exacerbated in the siRNA treatment group. However, neutrophil depletion (anti-Ly6G) or NETosis inhibition (DNase I) rescued HRG's protective effect in vivo, with reduced neurological deficits and HT (Fig. EV3). These findings suggested that the protective effect of HRG on HT was mediated by inhibiting neutrophils and NETosis.

## HRG inhibits neutrophil activation by downregulating pro-inflammatory signaling pathways

To elucidate the potential mechanism of action of HRG on neutrophil activity, we further analyzed the gene expression profiles of tPA- and tPA + HRG-treated neutrophils using RNA sequencing and proteomic profiling. Neutrophils were isolated for culture and subsequent sequencing, with >95% purity as identified by FACS analysis (Fig. EV4A). In total, 25,403 genes were detected in the tPA-treated vs. control group comparison. According to the adjusted p-value (padj) <0.05 and fold change >2, 90 DEGs were

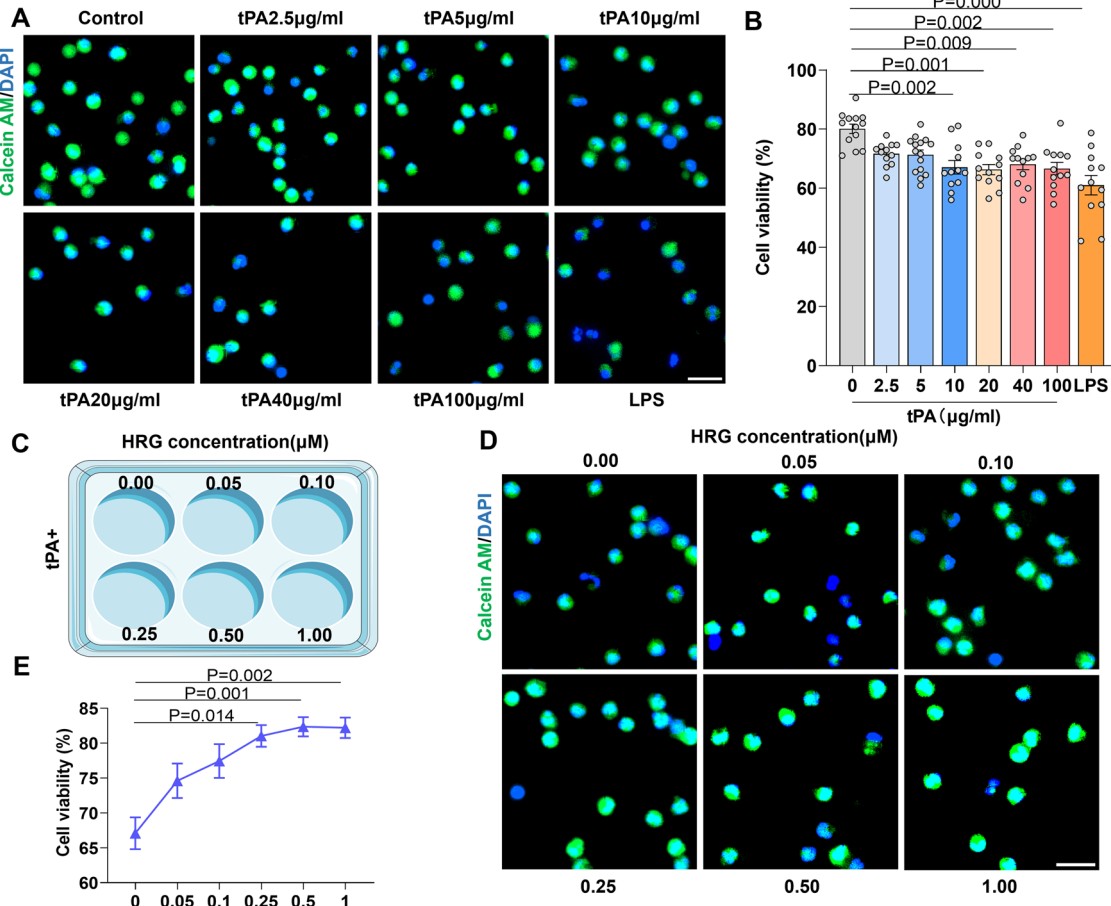

**Figure 3.   HRG increases the cell viability of purified human neutrophils after tPA treatment.**

(**A**) The neutrophils were labeled with calcein-AM (living cell, green) and Hoechst 33342 (nuclei, blue). Graded concentrations of tPA were added to the culture system with LPS (10 μg/ml) added as the positive control. After 18 h of culture, the viability of neutrophils was observed and calculated under a fluorescent microscope. Typical results from five independent experiments are shown. Scale bar = 20 μm. (**B**) Statistical analysis of the cell viability is shown ($n = 12$–15). (**C**) Schematic diagram illustrating the experimental design. (**D**) The culture system was treated with 10 μg/ml tPA and different concentrations of HRG (0.05 μM, 0.1 μM, 0.25 μM, 0.5 μM, 1 μM), and the viability of neutrophils was observed by calcein-AM staining. Scale bar = 20 μm. (**E**) The concentration-response curves of the effects of HRG on neutrophil mortality rate (0 μM, $n = 12$; 0.05 μM, $n = 13$; 0.1 μM, $n = 15$; 0.25 μM, $n = 12$; 0.5 μM, $n = 14$; 1 μM, $n = 13$). Data information: Data are presented as mean ± SEM. Two-tailed Kruskal–Wallis H test is used in (**B**, **E**). Source data are available online for this figure.

identified, including 83 upregulated and 7 downregulated genes (Fig. 7A). Similarly, 25,510 genes were detected in the tPA + HRG vs. tPA group comparison, including 80 DEGs, with 2 upregulated and 78 downregulated genes (Fig. 7B). Heatmap clustering analysis was employed to identify the top 20 DEGs between the two groups. The results showed significant upregulation of IL-1A, CCL3, CCL4, and other pro-inflammatory factors in the tPA group compared to the control group (Fig. 7C). Conversely, in the tPA + HRG vs. tPA groups, there was a notable downregulation of CCL3, CCL4, IL-6, and other pro-inflammatory factors (Fig. 7D). Pro-inflammatory cytokine expression in neutrophils was mainly upregulated after tPA treatment, whereas it was mostly downregulated after the addition of HRG. Furthermore, KEGG and GO analyses were employed for functional enrichment and pathway analyses of DEGs. The DEGs that were mostly upregulated by tPA treatment were significantly enriched in the NF-κB signaling pathway, along with the TNF signaling pathway, cytokine–cytokine receptor interaction, and other inflammation-related pathways (Fig. 7E).

In the tPA + HRG vs. tPA groups, the DEGs primarily downregulated under HRG treatment were significantly enriched in the NF-κB signaling pathway, cytokine–cytokine receptor interaction, and IL-17 signaling pathway (Fig. 7F). In addition, protein profiling of neutrophils identified a total of 183 DEPs (Fig. EV4A). KEGG classification revealed that the downregulated DEPs from the HRG treatment group were enriched for immune system pathways (Fig. EV4B). Furthermore, the KEGG pathway analysis indicated that the downregulated DEPs were enriched in pro-inflammatory signaling pathways, which include the NOD (nucleotide-binding oligomerization domain)-like receptor signaling pathway, the transcription factor nuclear factor kappa B (NF-κB) signaling pathway, and the interleukin-17 (IL-17) signaling pathway (Fig. EV4C). An integrated analysis of transcriptomic and proteomic data provided a comprehensive overview for the downregulation of neutrophil signaling pathways following HRG treatment. The combined analysis indicated that the correlated (Cor)-DEGs-DEPs were significantly enriched in the NOD-like

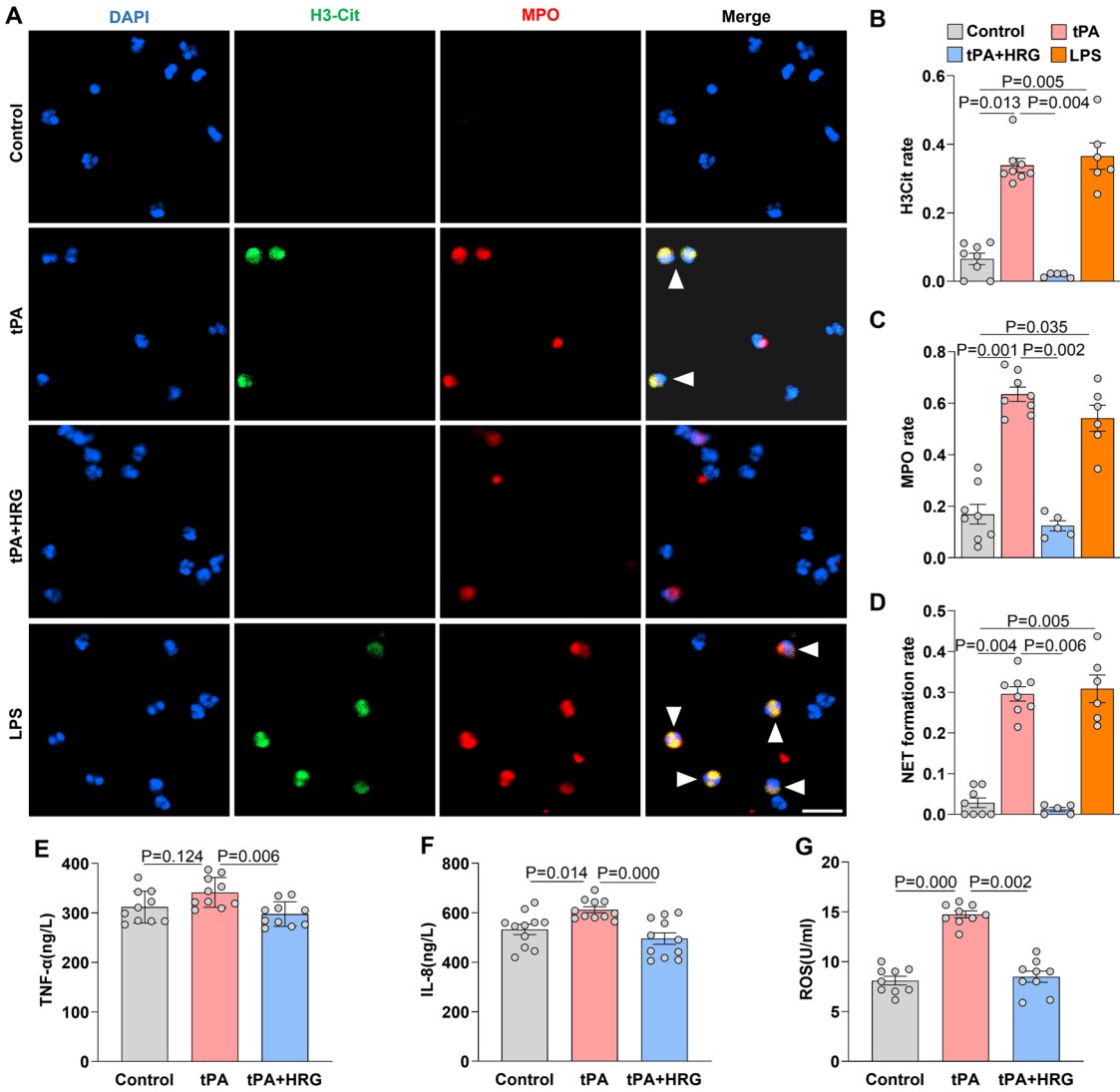

**Figure 4. HRG reduces NETosis and cytokine production of neutrophils after tPA stimulation.**

(A) Representative images of isolated peripheral blood neutrophils treated with tPA (10 μg/ml), tPA + HRG (1 μM), and LPS (10 μg/ml). Neutrophils were incubated for 2.5 h and co-stained with H3Cit (green), MPO (red), and DAPI (blue). White arrows indicate NET formation. Scale bar = 20 μm. (B–D) Statistical analysis and comparison of H3Cit, MPO, and NET formation rates in different groups ($n = 5$–8). The results shown are the mean ± SEM. (E–G) The concentration of TNF-α ($n = 10$), IL-8 ($n = 11$), and ROS ($n = 9$) in the supernatant of the culture system was quantified by ELISA. Data information: Data are presented as mean ± SEM; two-tailed Kruskal–Wallis H test is used in (B–G). Source data are available online for this figure.

receptor signaling pathway, at both mRNA and protein levels (Fig. EV4D). In addition, GO analysis of the DEGs indicated that the biological processes enriched in the upregulated genes of tPA vs. control group were related to cellular inflammatory responses, positive regulation of cell migration, and positive regulation of leukocyte migration. Meanwhile, the inflammatory response, cell migration, and cell motility were downregulated in the HRG treatment group (Fig. EV5A,B). GO analysis of the proteomic data also supported downregulation of chemotaxis and immune responses following HRG treatment (Fig. EV5C). These findings indicate that HRG exerts an anti-inflammatory effect on tPA-induced neutrophil activation, affecting the inflammatory responses and cell migration, most likely by regulating NOD-like receptor signaling and downstream NF-κB activation.

## Discussion

In the present study, proteomic analyses identified upregulation of HRG plasma levels following intravenous tPA infusion in patients with AIS. Elevated HRG levels were further verified using ELISA in a validation cohort of AIS patients with or without tPA treatment. AIS per se did not induce significant changes in plasma HRG levels, and neither did AIS disease progression during the first 6.5 h after symptom onset. To our surprise, insufficient HRG increases at 1 h post-tPA infusion correlated with increased risk of developing HT within 24 h of tPA treatment, which appears unexpected based on previous reports regarding its physio-pathological role in thrombosis (Malik et al, 2021; Shigekiyo et al, 1998; Vu et al, 2015). In vitro experiments demonstrated that HRG can effectively reduce neutrophil

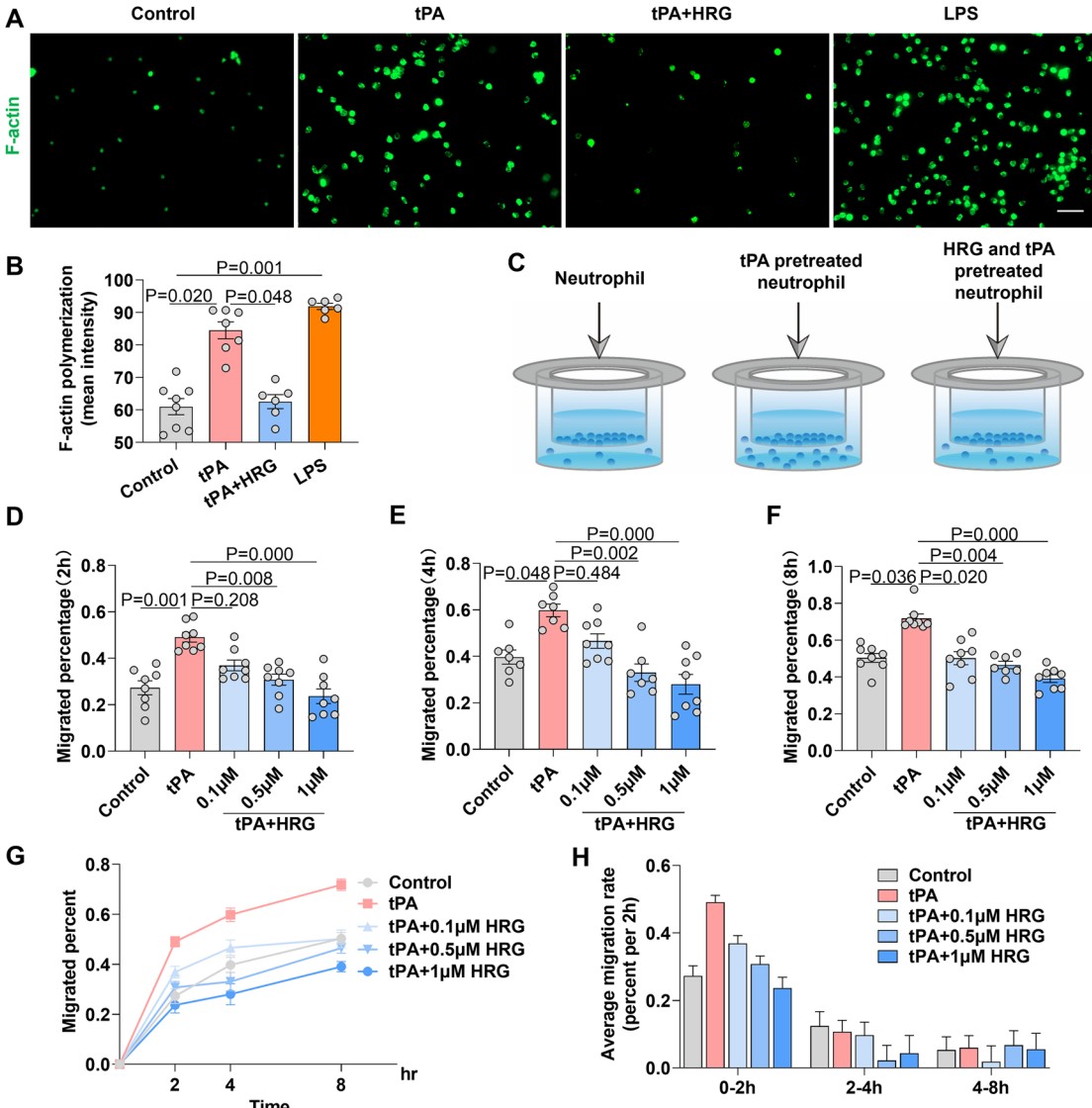

**Figure 5. HRG inhibits neutrophil polymerization of F-actin and migration across the BBB in vitro.**

(A) The polymerization of F-actin in neutrophils was observed under a fluorescent microscope by staining with phalloidin (cytoskeleton, green) and Hoechst 33342 (nuclei, blue). The cells were observed at 2.5 h after stimulation with tPA (10 µg/ml), tPA (10 µg/ml) + HRG (0.5 µM), and LPS (10 µg/ml). Scale bar = 20 µm. (B) Quantitative analysis of F-actin immunofluorescence intensity under different conditions (*n* = 6–8). (C) Schematic diagram illustrating the experimental design. Transwell membrane inserts were coated with collagen IV and fibronectin, and hCMEC/D3 cells were seeded onto them followed by 4 h of hypoxia-glucose deprivation. Neutrophils pretreated with different factors (tPA: 10 µg/ml, HRG: 0.1 µM, HRG: 0.5 µM, and HRG: 1 µM) for 2 h were added to the upper chamber. After 2 h (D) (*n* = 8), 4 h (E) (*n* = 7–8), and 8 h (F) (*n* = 7–8) of culture, the number of neutrophils in the lower chamber was counted. Migration percentage = the number of neutrophils in the lower chamber/the total number of neutrophils added to the upper chamber. (G) The line graph summarizes the migrated percentage of neutrophils at 2 h, 4 h, and 8 h (*n* = 6–8). (H) The average migration rate of neutrophils during the 0–2 h, 2–4 h, and 4–8 h of the transwell assay. Data information: Data are presented as mean ± SEM; two-tailed Kruskal–Wallis H test is used in (B, D–F). Source data are available online for this figure.

mortality rate, inhibit their ability to cross the BBB, suppress NETosis and ROS, and decrease the release of inflammatory factors such as TNF-α and IL-8. Administration of HRG ameliorated delayed thrombolysis induced HT in MCAO mice treated with tPA at 5 h post stroke onset, suggesting that HRG can reduce the risk of tPA-related HT and may extend the time window for tPA administration in AIS. These results indicate that plasma HRG levels play a key role in fine-tuning immune and thrombolytic homeostasis.

The mechanism of tPA-associated HT is complex. Tissue plasminogen activator is secreted endogenously by endothelial cells

and promotes the conversion of plasminogen into plasmin (Collen and Lijnen, 1991). Accordingly, tPA achieves fibrinolysis by converting inactive fibrin-bound plasminogen into plasmin, which in turn breaks down fibrin into split products. An early increase in fibrin degradation products has been reported to be associated with a substantially increased risk of HT and hematoma expansion (Trouillas et al, 2004). In addition, the administration of tPA further enhances neuroinflammation following ischemic stroke, resulting in BBB disruption and allowing infiltration of peripheral immune cells into the brain (Liu et al, 2023; Ma et al, 2021; Shi et al, 2021; Zubair and Sheth, 2023).

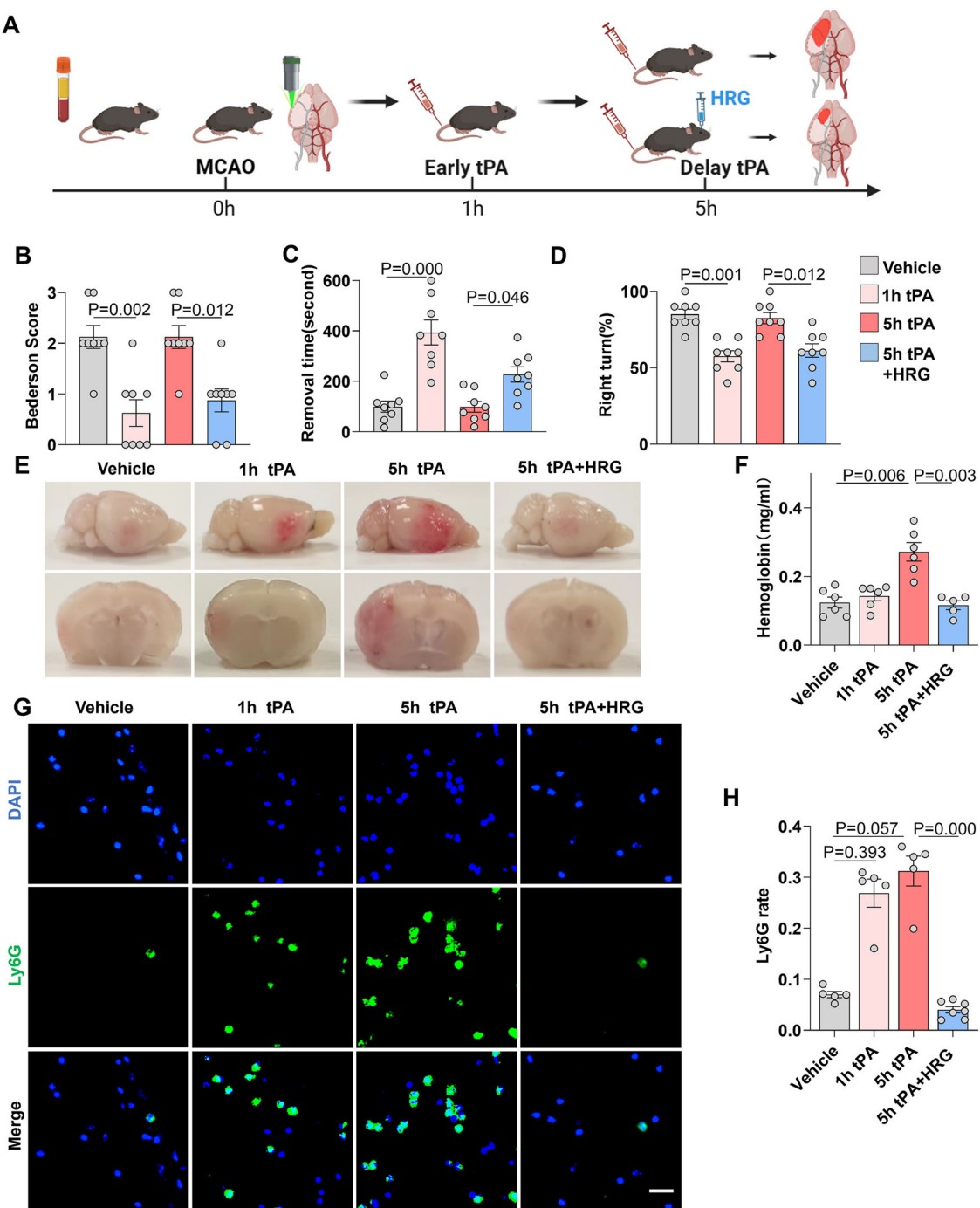

**Figure 6. HRG alleviates hemorrhagic transformation and neutrophil infiltration in tPA-treated MCAO mice.**

(A) Flow chart illustrating the experimental design. An animal model of photothrombotic MCAO was performed, with blood collected at 1 h and 5 h for HRG determination. In addition, tPA was administered separately at 1 h and 5 h, indicating early and delayed thrombolysis. HRG in combination with tPA was also administered at 5 h after MCAO. Neurological deficit, HT, and neutrophil infiltration were evaluated. (B) Neurological deficits were measured by assessing the Bederson score at 24 h after stroke (n = 8). Sensorimotor function was measured using the adhesive removal test (C) (n = 8) and corner test (n = 8) (D). (E) Representative images of the dorsal surface and a coronal section showing cerebral hemorrhage 24 h after stroke in mice treated with vehicle, tPA (1 h after stroke), tPA (5 h after stroke), or tPA + HRG (5 h after stroke). (F) Quantification of cerebral hemorrhage by spectrophotometric hemoglobin assay (n = 6). (G) Representative images of immunofluorescence staining of infiltrating neutrophils (Ly6G, green) in brain lesions. Scale bar = 20 μm. (H) The graph depicts the proportion of Ly6G-positive cells (n = 5–7). The same data are also partially used in Fig. EV3B. Data information: The results are shown as mean ± SEM. Two-tailed Kruskal–Wallis H test is used in (B–D, F, H). Source data are available online for this figure.

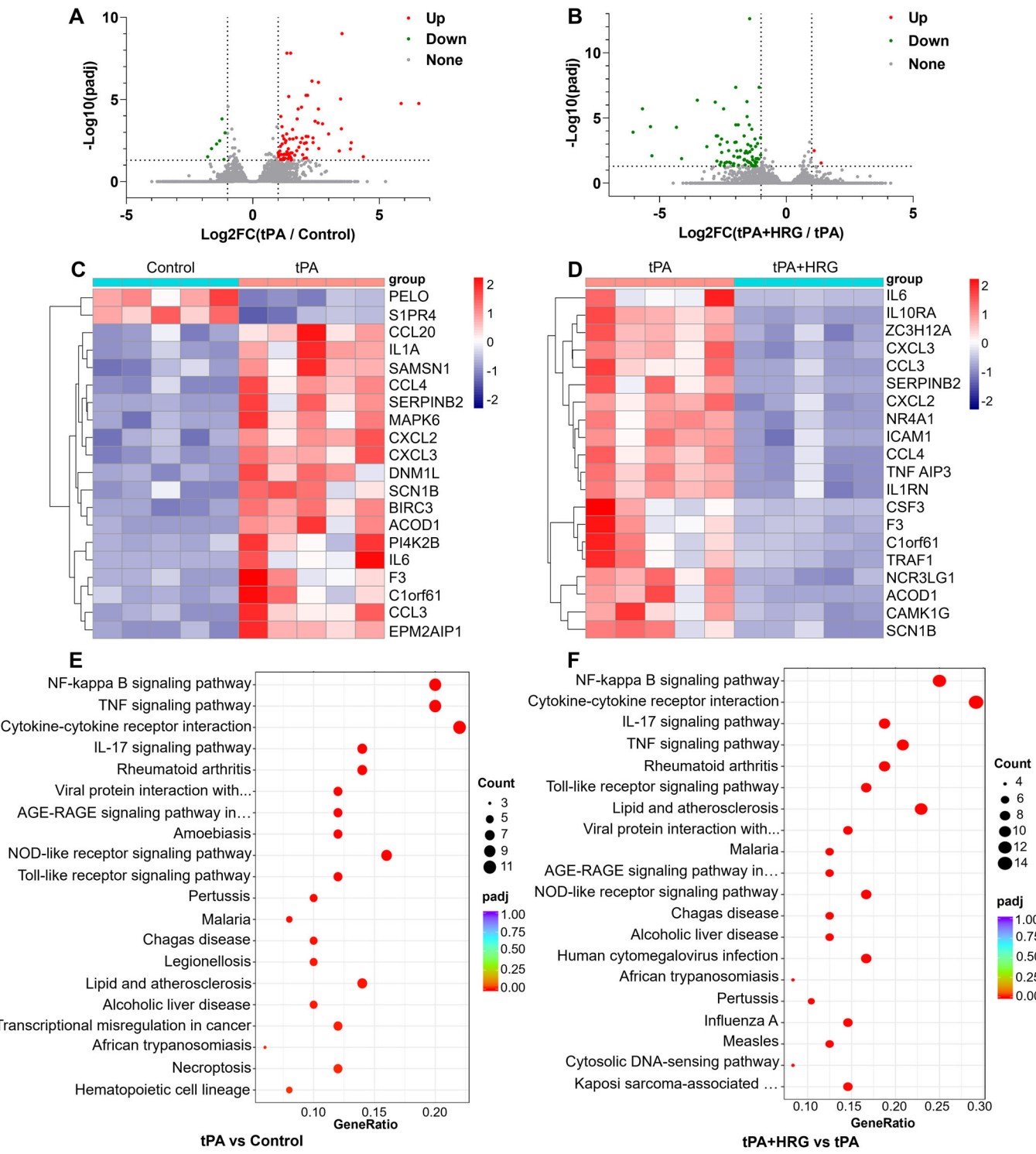

**Figure 7. RNA sequencing and KEGG analysis for neutrophil gene expression under HRG treatment with tPA.**

(A, B) The gene expression profiles of neutrophils under vehicle, tPA, and tPA + HRG treatments were analyzed by RNA sequencing. We performed a hypergeometric test and used Benjamini-Hochberg (BH) correction for padj. DEGs between each group were identified as the padj <0.05 and fold change >2. The volcano plots for group tPA vs. control (A) and group tPA + HRG vs. tPA (B) are shown. (C) Cluster analysis chart of the top 20 identified DEGs for the tPA vs. control group. (D) Cluster analysis chart of the top 20 identified DEGs for the tPA + HRG vs. tPA group. Upregulated genes are shown in red. Downregulated genes are shown in blue. The dendrogram depicts the hierarchical relationship among genes. (E) KEGG pathway analysis for DEGs in the tPA vs. control group. (F) KEGG pathway analysis for DEGs in the tPA + HRG vs. tPA group (n = 5). We performed KEGG functional enrichment for DEGs by clusterProfiler R package (3.8.1). For statistical methods, we used hypergeometric test and BH correction for padj.

These fast-reacting immune cells, known as neutrophils, play a crucial role in HT, with increased matrix metalloproteinase-9 (MMP-9), neutrophil elastase, and MPO (Carbone et al, 2015; Wang et al, 2021; Zhang et al, 2021). Therefore, high-throughput quantitative characterization of circulating proteins in patients with or without tPA treatment may provide pathophysiological insights into understanding the effective response to tPA treatment, predicting potential biomarkers for HT, and identifying therapeutic targets for relieving side effects. Elevation of HRG was identified and emerged as playing a central role in bridging immunity and fibrinolysis.

Elevation of HRG in plasma may originate from increased synthesis in the liver, the release from platelet α-granules, or impaired degradation. HRG is synthesized in the liver and is present in plasma with high-affinity binding to a number of ligands (Koide et al, 1986; Leung, 1986; Lijnen et al, 1980; Poon et al, 2010), which could be increased by altered plasma components upon thrombolysis. HRG is also stored in human platelets and released following stimulation (Leung et al, 1983). Declined deposition of HRG on immunothrombi and impaired degradation of HRG by thrombin could also contribute to the changes in HRG levels in plasma (Wake et al, 2016). To further clarify the origin of HRG following tPA treatment, cells from liver, peripheral blood, and various tissues of mice were collected for HRG determination. HRG mRNA expression in the liver increased as early as 1 h post-tPA treatment, and slightly decreased at 4 h post-tPA treatment (Appendix Fig. S5A). However, we were unable to detect HRG mRNA in immune tissues including the spleen, thymus, lymph node, bone marrow, and peripheral blood leukocytes in C57BL/6 mice, which is consistent with a previous report (Hulett and Parish, 2000) (Appendix Fig. S5B). In addition, it has been shown that HRG is present in platelets and released upon thrombin stimulation in human blood (Leung et al, 1983). To determine if platelet-derived HRG contributed to increases in plasma HRG levels following tPA treatment, platelet-rich plasma (PRP) was collected from human peripheral blood and stimulated with tPA in vitro. After 40 min of incubation, PRP, platelet-poor plasma (PPP) and platelet were collected for HRG determination by Western blot analysis (Appendix Fig. S5C). However, no alterations in HRG were detected in PRP, PPP, and platelet after tPA stimulation (Appendix Fig. S5D). Therefore, it indicated that liver primarily contributed to the origin of HRG in plasma after tPA stimulation. The significance and mechanism of HRG elevation in affecting HT was further addressed in our study.

Data from the current study uncovered that HRG possesses an immune modulatory role regulating neutrophil function in AIS. Of note, HRG has been reported to suppress sepsis and tumor metastasis, inhibit hyperinflammation in acute pancreatitis, and relieve liver ischemia/reperfusion injury (Guo et al, 2021; Kuroda et al, 2021; Terao et al, 2018; Tomonobu et al, 2022; Yin et al, 2023). Our study revealed that HRG effectively suppresses neutrophil infiltration and immune activation following thrombolysis. In this study, RNA sequencing and proteomic analysis was used to explore the possible signaling pathways behind the effects of HRG on neutrophils. HRG significantly suppressed the activation of the NOD-like receptor signaling pathway and NF-κB-dominant inflammatory pathways in neutrophils following tPA treatment, which is consistent with previous findings in endothelial cells (Gao et al, 2019). These results contribute to a more comprehensive understanding of the immunomodulatory functions of HRG.

In addition to its effect on neutrophil modulation, HRG has been shown to interact with components of both the coagulation and fibrinolytic systems, playing an important role in homeostasis (Poon et al, 2011). Recent studies indicated that a high concentration of HRG

may reduce contact activation and thrombosis without affecting hemostasis (Malik et al, 2023; Schmaier, 2023). HRG was first shown to bind strongly to plasminogen via its lysine-binding site, thereby affecting plasminogen activation (Lijnen et al, 1980). In addition, it has been suggested that HRG can not only bind to fibrinogen with a high affinity to prolong thrombin time (Leung, 1986) but also inhibit fibrinogen-dependent plasminogen activation (Borza et al, 2004). These studies provided evidence that HRG displays both anti-coagulative and anti-fibrinolytic effects, which led to the notion that the dynamic variation in HRG levels in plasma may serve as a bidirectional valve in regulating coagulative and fibrinolytic homeostasis, especially in complex cascades following thrombolysis. Early elevated HRG levels (within 1–3 h after tPA administration) in the plasma of patients with AIS following tPA treatment may contribute to the antithrombotic effect observed during thrombolysis. However, with the gradual increase in fibrinolytic products and neutrophil mobilization following infusion, sustained higher levels of HRG (5–24 h after tPA treatment) contribute to anti-fibrinolytic and anti-inflammatory processes to counterbalance the hemorrhagic-prone state, resulting in reduced bleeding risk.

Multiple studies have explored possible biomarkers of HT in ischemic stroke. However, differences in the timing of blood sampling limits comparisons across studies and our results. Studies with only pre-tPA samples revealed that higher levels of MMP-9 (Montaner et al, 2003), serum cellular-fibronectin (c-Fn) (Castellanos et al, 2004) and S100B (Foerch et al, 2007) at baseline are associated with HT in patients treated with tPA (Castellanos et al, 2007). Biomarker studies from pre- vs. post-tPA comparisons were mostly collected at baseline and at 24 h post-tPA treatment, among which MMP-9 (Inzitari et al, 2013), C-reactive protein (CRP) (Karlinski et al, 2014) and IL-10 (Gori et al, 2017) were related to HT prediction. Only one study compared samples collected at baseline and within 3 h post-tPA treatment, which showed the presence of the HRG protein precursor among degraded protein fragments post-tPA administration (Ning et al, 2010). Ning et al reported differences in protein expression before and 3 h after thrombolysis in patients with stroke. This study demonstrated that t-PA treatment changed the plasma degradomic profiles in patients with acute stroke, and HRG precursor protein was found among the degraded protein fragments in degradomic fractions post-tPA administration. However, no information on quantitative protein variations were reported due to limitations in proteomic quantification (Ning et al, 2010).

Some limitations of this study should be considered. The number of AIS patients with symptomatic HT in the validation cohort is low, which raises concerns about the translational predictive value of HRG levels in clinical practice. The impaired HRG elevation observed in patients with HT following thrombolysis requires further validation in a large prospective cohort study. Hepatic function and platelet counts of AIS patients in our discovery and validation cohort were within the normal range, and none of the individual parameters in liver function test were associated with impaired elevation of HRG upon tPA infusion. The underlying mechanisms for the lack of sufficient increases in plasma HRG levels following intravenous tPA infusion remain to be fully defined. Moreover, with the increased application of tenecteplase, which possesses a longer plasma half-life and achieves thrombolysis as a single bolus injection, variations in HRG levels should be determined in a tenecteplase-treated patient cohort.

In summary, this study demonstrated that increased HRG levels after tPA treatment can regulate neutrophil immune activity and inhibit neutrophil infiltration and excessive immune activation, thereby reducing the risk of tPA-related HT and extending the

thrombolytic time window. HRG may serve as a potential target for predicting and managing HT, although further investigation is required to elucidate its underlying mechanism.

## Methods

### Reagents and tools table

| Reagent/Resource | Reference or Source | Identifier or Catalog Number |
|---|---|---|
| **Experimental Models** | | |
| Plasma | This study | N/A |
| Neutrophils | This study | N/A |
| C57BL6/J (M. musculus) | Vital River Corporation | N/A |
| AML12 cells | Procell Life Science & Technology Co, Ltd | CL-0602 |
| hCMECs | Zhongqiao Xinzhou Biotechnology | ZQ0961 |
| **Recombinant DNA** | | |
| N/A | N/A | N/A |
| **Antibodies** | | |
| Rabbit anti-Histone H3 | Abcam | ab219407 |
| Mouse anti-Myeloperoxidase | Invitrogen | MA1-80878 |
| Rabbit anti-CLEC1A | Proteintech | 13394-1-AP |
| Rabbit anti-CLEC1B | Proteintech | 12814-1-AP |
| Alexa-Fluor 488 donkey anti-rabbit secondary antibody | Invitrogen | A-21206 |
| Alexa-Fluor 594 donkey anti-mouse secondary antibody | Invitrogen | A-32744 |
| Alexa-Fluor 488-conjugated phalloidin | Invitrogen | A12379 |
| Anti-mouse Ly6G | Bio X Cell | BE0075-1 |
| Rat anti-Ly6G | Biolegend | 127605 |
| Alexa-Fluor 488-conjugated secondary antibody | Invitrogen | A-21208 |
| Rabbit anti-HRG | Proteintech | 26252-1-AP |
| **Oligonucleotides and other sequence-based reagents** | | |
| HRG siRNA | This study | Table EV6 |
| PCR primer (GAPDH) | This study | Table EV6 |
| PCR primer (HRG) | This study | Table EV6 |
| **Chemicals, Enzymes, and other reagents** | | |
| RPMI-1640 medium | Procell | PM150110 |
| Hoechst 33342 | Solarbio | C0031 |
| Calcein-AM | Glpbio | GC34061 |
| EGM-2 MV medium | Lonza | Cc-3162 |
| LPS | Solarbio | L8880 |
| Rose bengal | Sigma-Aldrich | 330000 |
| D-Glucose | Sigma-Aldrich | 158968 |

| Reagent/Resource | Reference or Source | Identifier or Catalog Number |
|---|---|---|
| tPA | Boehringer Ingelheim | N/A |
| Recombinant histidine-rich glycoprotein | Sino Biological | 10836-H08H |
| DNase I | Sigma-Aldrich | D5025 |
| Red blood cell lysate | Solarbio | R1010 |
| EntransterTM in vivo | Engreen Biosystem Co, Ltd | 18668-11-1 |
| TRIzol reagent | Invitrogen | 15596026CN |
| PerfectStart SYBR Green qPCR super mix | Transgen | AQ601 |
| **Software** | | |
| MaxQuant software | https://www.maxquant.org | |
| MSstats software | https://bioconductor.org | |
| DAVID software | https://david.ncifcrf.gov | |
| STRING database | http://string-db.org | |
| Hisat2 software v2.0.5 | http://daehwankimlab.github.io/hisat2 | |
| FeatureCounts v1.5.0-p3 | https://sourceforge.net | |
| DESeq2 R package v1.20.0 | https://bioconductor.org/packages/release/bioc/html/DESeq2.html | |
| ClusterProfiler v3.8.1 | https://www.bioconductor.org/packages/release/bioc/html/clusterProfiler. | |
| GraphPad Prism v8.3.0 | https://www.graphpad.com | |
| Image J | https://imagej.nih.gov/ij/index.html | |
| FlowJo v10.8.1 | https://www.flowjo.com | |
| SPSS 22.0 | https://www.ibm.com/spss | |
| **Other** | | |
| Bradford assay kit | Bio-Rad | |
| UltiMate 3000UHPLC liquid chromatograph | Thermo Fisher Scientific | |
| Q-Exactive HF X | Thermo Fisher Scientific | |
| Human neutrophils density gradient separation kit | Solarbio | |
| Biotek Cytation 5 | Biotek | |
| HRG ELISA kit | Sino Biological | |
| IL-8 ELISA kit | MEIMIAN | |
| TNF-α ELISA kit | MEIMIAN | |
| ROS kit | KeyGENBioTECH | |
| Transwell membrane inserts | Corning Life Science | |
| Hemoglobin content assay kit | Solarbio | |
| Bio-Rad CFX96 real-time system | Bio-Rad | |

## Patient cohort and plasma collection

From June 2017 to July 2019, ten AIS patients who received tPA treatment were enrolled in the discovery cohort for plasma collection and proteomic analyses. Blood samples were drawn in the emergency room prior to the initiation and 1 h after the completion of tPA infusion.

In the validation cohort, 104 patients with AIS were enrolled for ELISA detection from March 2020 to May 2024. Fifty-three healthy volunteers served in the healthy control (HC) group. Sixty-two patients with AIS who received tPA treatment were enrolled in the AIS + tPA group, whereas 42 patients who declined or were not eligible for thrombolysis were enrolled as the AIS w/o tPA group. Plasma samples were collected at baseline, prior to the initiation of and at 1 h after the completion of tPA infusion. For patients in the AIS w/o tPA group, blood samples were drawn at 3.5 h and 6.5 h post-stroke symptom onset. This study was approved by the Tianjin Medical University General Hospital Ethics Committee (IRB2020-YX-214-01). Informed consent was obtained from all participants, and all experiments were conducted in accordance with Chinese laws and institutional guidelines. The experiments conformed to the principle of WMA Declaration of Helsinki and the Department of Health and Human Services Belmont Report. Stroke severity was assessed using the National Institutes of Health Stroke Scale (NIHSS) at baseline. Baseline data for these patients are shown in Fig. 2A and Table EV1.

## Protein extraction, enrichment, and digestion

For neutrophils, the samples were taken and an appropriate amount of protein lysate containing SDS (sodium dodecyl sulfate) and a final concentration of 1XCocktail (protease inhibitor) containing EDTA (diethylamine tetraacetic acid) were added. The cells were lysed by sonication, centrifuged at $25,000 \times g$ at 4 °C for 15 min, and the supernatant was collected. Then for plasma and cells, Acetone was added to the plasma samples, and the mixture was centrifuged to separate the supernatant. The resultant pellet was air dried, and 200 μl lysis buffer containing Urea, Thiourea, SDS and Tris was added to redissolve the pellet. Subsequently, dithiothreitol and iodoacetamide were added for alkylation at room temperature in the dark, and the supernatant protein concentration was determined using a Bradford assay kit (Bio-Rad, Hercules, CA, USA). The diluted plasma protein mixtures were loaded onto and passed through the SPE C18 column at a rate of 1 drop per second. Subsequently, the non-specific binding proteins were washed, and the enriched proteins were eluted. Each sample was diluted with the appropriate amount of protein solution and mixed with the appropriate amount of sample buffer for electrophoresis. Thereafter, trypsin was added to the protein solution at a ratio of 1:20 (w/w). The mixture was incubated for 14–16 h at 37 °C.

## High pH reversed-phase HPLC separation

Equal amounts of peptides from each sample were pooled and diluted with 2 ml mobile phase A (5% ACN, pH 9.8) for injection. A Shimadzu LC-20AB HPLC system coupled with a Gemini high pH C18 column (5 μm, 4.6 × 250 mm) was used. The sample was subjected to the column and eluted at a flow rate of 1 ml/min using the following gradient: 5% mobile phase B (95% ACN, pH 9.8) for

10 min, 5–35% mobile phase B for 40 min, 35–95% mobile phase B for 1 min, flow phase B for 3 min, and finally 5% mobile phase B was equilibrated for 10 min. Eluates were collected every 1 min. Finally, all fractions were combined into 10 fractions, which were then frozen and dried.

## Data-dependent acquisition (DDA) and data-independent acquisition (DIA) analysis by nano-liquid chromatography tandem mass spectrometry (LC-MS/MS)

The dried peptide samples were reconstituted with mobile phase A (2% ACN, 0.1% FA) and centrifuged, and the supernatant was collected for injection. For plasma, the separation was performed using a Thermo UltiMate 3000UHPLC liquid chromatograph, and for neutrophils, separation was performed by Bruker nanoElute. The sample was first enriched in the trap column, desalted, and then entered into a tandem self-packed C18 column. Subsequently, it was separated by the effective gradient, The nanoliter LC separation end was directly connected to the mass spectrometer and detected according to the following parameters: For plasma mass spectrometry parameters were set as follows for the DDA mode: ion source voltage 1.9 kV; MS scan range, 350–1500 $m/z$; MS resolution 120,000, maximal injection time (MIT) 50 ms; MS/MS collision-type HCD, collision energy NCE 28; MS/MS resolution 30,000, MIT 100 ms, dynamic exclusion duration 30 s. The starting $m/z$ value for MS/MS was fixed at 100. The precursor for the MS/MS scan satisfied the following criteria: charge range 2+ to 6+ and the top 20 precursors with intensity over 2E4. The AGC used were MS 3E6 and MS/MS 1E5. For neutrophils, the mass spectrometry parameters were set as follows for DDA mode: ion source voltage was set to 1.6 kV, MS1 mass spectrometer scanning range was 100–1700 $m/z$; ion mobility range was 0.6–1.60 V.S/cm²; MS2 mass spectrometer scanning range was 100–1700 $m/z$; The precursor ion screening conditions for MS2 fragmentation: charge 0 to 5+, and the first 10 precursor ions with the peak intensity exceeding 10,000 and the peak intensity above 2500 can be detected. The MS1 cumulative scan time was 100 ms; MS2 cumulative scan time was 100 ms; The ion fragmentation mode was CID, and the fragment ions were detected in TOF. The dynamic exclusion time was set to 30 s. For the DIA analysis used for plasma, the same nano-LC system and gradient were used as in the DDA analysis. The main settings were as follows: ion source voltage 1.9–2 kV; MS scan range, 400–1250 $m/z$; MS resolution, 120,000, MIT 50 ms; 400–1250 $m/z$, equally divided into 45 continuous window MS/MS scans. The MS/MS collision-type HCD MIT was in auto mode. Fragment ions were scanned in Orbitrap, MS/MS resolution 30,000, collision energy was in distributed mode: 22.5, 25, and 27.5, and the AGC was 1E6. For DIA analysis used for neutrophils, the main settings were: ion source voltage was set to 1.6 kV, ion mobility range was 0.6–1.60 V.S/cm²; MS1 mass spectrometer scanning range was 302–1077 $m/z$ and the peak intensity above 2500 can be detected; The 302–1077 $m/z$ was divided into 4 steps, and each step was divided into 8 Windows. A total of 32 Windows was used for continuous window fragmentation and information collection. The fragmentation mode was CID, the fragmentation energy was 10 eV, and the mass width of each window was 25. The cycle time of a DIA scan was 3.3 s.

## Protein identification, quantitative analysis, and bioinformatics analysis

MaxQuant software (version 1.5.3.30) was used to identify the DDA data as a spectral library for subsequent DIA analysis. The identified peptides satisfying FDR ≤ 1% were used to construct the final spectral library. The parameters were configured as follows: enzyme: trypsin; minimal peptide length: 7; PSM-level FDR and protein FDR: 0.01; fixed modifications: carbamidomethyl (C); variable modifications: oxidation (M); acetyl (protein N-term); Gln- > pyro-Glu (N-term Q); deamidated (NQ). DIA data were analyzed using iRT peptides for retention time calibration. Then, based on the target-decoy model applicable to SWATH-MS, a false-positive control was performed with an FDR of 1%; therefore, significant quantitative results were obtained. Finally, the MSstats software package was used to evaluate statistically significant differences in the proteome. For plasma, differential protein screening was performed based on a fold change >2, with a $p$-value < 0.05 as the criterion for a significant difference. And for neutrophils, differential protein screening was performed based on the fold change >1.5 and $P$ value < 0.05 as the criterion for the significant difference.

The annotation function of the proteins was analyzed by Gene Ontology (GO) using the online DAVID software. The Kyoto Encyclopedia of Genes and Genomes (KEGG) pathway database was used for pathway analysis of the identified plasma proteins. Protein–protein interaction (PPI) network analysis was performed using the STRING database (http://string-db.org).

## Isolation and culture of human peripheral blood neutrophils

Peripheral blood was collected from AIS patients within the thrombolytic time window. Neutrophils were extracted using the density gradient separation kit (P9040, Solarbio, Beijing, China). Briefly, whole blood was centrifuged for 10 min, and the plasma was discarded. Then, the separation reagent and blood cells were sequentially added to a 15 ml tube and centrifuged at $800 \times g$ for 30 min. The neutrophil layer was retained according to the manufacturer's instructions and resuspended with red blood cell lysate (R1010, Solarbio, Beijing, China) at 4 °C for 15 min. Cells were centrifuged and resuspended with RPMI-1640 medium (PM150110, Procell, Wuhan, China) and cultured at 37 °C. The purity of the isolated neutrophils was determined to be >98% by fluorescence-activated cell sorting analysis (Appendix Fig. S6A).

### Neutrophil viability assay

The neutrophil suspension was aliquoted to a 96-well plate (Corning Life Science, Lowell, MA) in a volume of 100 μl ($5 \times 10^5$ cells per ml) with one of the following reagents: PBS, tPA (2.5–100 μg/ml), tPA + HRG (0.05–1 μM), or LPS (10 μg/ml; Sigma-Aldrich). Anti-CLEC1A antibodies (1:100, 13394-1-AP, Proteintech, Wuhan, China) and anti-CLEC1B antibodies (1:100, 12814-1-AP, Proteintech, Wuhan, China) were employed to block the receptors prior to HRG treatment. After stimulation at 37 °C and 5% $CO_2$ for 4 h, 8 h, 12 h, 18 h, and 24 h, neutrophils were stained with Hoechst 33342 (C0031, Solarbio, Beijing, China) and calcein-AM (GC34061, Glpbio, California, USA), after which the neutrophils were scanned. Calcein-negative neutrophils were

counted using Biotek Cytation 5. The total calcein-AM fluorescence intensity of the image was calculated using image J and divided by the number of cells to obtain the average calcein-AM fluorescence intensity of each cell in that field. Neutrophil viability = average calcein-AM fluorescence intensity of each cell/average calcein-AM fluorescence intensity of the positive control.

### In vitro NETosis assay

Neutrophils were isolated, seeded at a density of $5 \times 10^5$ cells/ml in 96-well plates (Corning Life Science, Lowell, MA), and treated with PBS, tPA (10 μg/ml), tPA (10 μg/ml) + HRG at 0.05 μM, 0.1 μM, 0.25 μM, 0.5 μM, or 1 μM, or with LPS (10 μg/ml) at 37 °C for 2.5 h. Anti-CLEC1A antibodies (1:100) and anti-CLEC1B antibodies (1:100) were added prior to HRG treatment (0.5 μM), and co-cultured with neutrophils at 37 °C for 2.5 h. Then, neutrophils were fixed in 2% paraformaldehyde, blocked with 3% bovine serum albumin, and incubated with rabbit anti-Histone H3 (H3Cit; 1:300; ab219407, Abcam, MA, USA) and mouse anti-Myeloperoxidase (MPO; 1:300; MA1-80878, Invitrogen, USA), followed by incubation with Alexa-Fluor 488 donkey anti-rabbit secondary antibody (1:1000; A-21206, Invitrogen, CA, USA) and Alexa-Fluor 594 donkey anti-mouse secondary antibody (1:1000; A-32744, Invitrogen, CA, USA). Hoechst 33342 (Solarbio, Beijing, China) was used for nuclear staining. NETosis was observed by fluorescence microscopy (Biotek Cytation 5, USA) and analyzed by ImageJ.

### ELISA measurement of HRG in human/mouse plasma, as well as IL-8, TNF-α, and ROS in the supernatant

Plasma samples were collected from whole blood and HRG levels were measured using a human HRG ELISA kit (KIT10836, Sino Biological, Beijing, China). Absorbance was measured at 450 nm. Levels of IL-8 and TNF-α in neutrophil culture supernatants were measured with a human IL-8 (IL-8/CXCL8) ELISA kit (MM-50812H1, MEIMIAN, Suzhou, China) and human TNF-α ELISA kit (MM-50812H1, MEIMIAN, Suzhou, China), respectively. Absorbance was measured at 450 nm. ROS levels were detected using a ROS kit (KGT010, KeyGENBioTECH, Nanjing, China) according to the manufacturer's protocol.

### Determination of total cellular F-actin

Freshly isolated neutrophils were aliquoted into a 96-well plate (Corning Life Science, Lowell, MA) ($5 \times 10^5$ cells per ml) and mixed with each of the following reagents: PBS, tPA (10 μg/ml), tPA (10 μg/ml) + HRG at 0.05 μM, 0.1 μM, 0.25 μM, 0.5 μM, or 1 μM, or LPS (10 μg/ml). After incubation at 37 °C for 2.5 h, neutrophils were fixed in 2% paraformaldehyde, treated with Triton X-100, and blocked with 3% bovine serum albumin. They were then stained with Alexa-Fluor 488-conjugated phalloidin (Thermo Fisher Scientific, Waltham, MA, USA) for 60 min. F-actin levels were determined by fluorescence microscopy (Biotek Cytation 5, USA). All images were captured using identical camera settings (time of exposure, brightness, contrast, and sharpness), and an appropriate white balance was set according to the fluorescence filter. Images were acquired and analyzed using ImageJ. The mean fluorescence density was determined from a linear measurement of individual cell fluorescence in randomly chosen fields on each slide.

## In vitro model of the BBB and permeability assay

Human brain microvascular endothelial cells (hCMECs; Zhongqiao Xinzhou Biotechnology, Shanghai, China) between passages three and eight were grown in EGM-2 MV medium (Lonza, Walkersville, MD, USA). Transwell membrane inserts (3 μm pores, 11 mm in diameter; Corning Life Science, Lowell, MA, USA) were coated with collagen IV (50 mg/ml) and fibronectin (30 mg/ml). The hCMECs (density, $2 \times 10^5$ cells) were seeded onto the upper side of the insert and cultured in complete medium composed of EGM-2 MV medium (Lonza, Walkersville, MD) with 10% fetal bovine serum at 37 °C for 3–4 days. When the monolayer of cells became confluent, the cell culture medium was replaced with glucose-free medium and exposed to hypoxia (95% nitrogen and 5% $CO_2$) at 37 °C for 4 h. Then neutrophils pretreated with PBS, tPA, or (tPA + HRG at 0.1 μM, 0.5 μM or 1 μM) for 2 h were added to the upper chamber after counting. The neutrophil count in the lower chamber of each group was observed at 2 h, 4 h, and 8 h after incubation. The cell migration rates were calculated as follows: cell counts in the lower chamber/initial cells added to the upper chamber.

## Photothrombotic stroke model and tPA thrombolysis

Animal experiments were performed in accordance with the National Institutes of Health Guide for the Care and Use of Laboratory Animals following an institute-approved protocol. Male C57BL/6 mice aged 8–10 weeks were acquired from Vital River Corporation (Beijing, China), and were used for photothrombotic stroke models. All mice were housed under specific pathogen-free conditions with a 12-h light/dark cycle, and given free access to food and water. The experiment was performed in compliance with ARRIVE and RIGOR criteria. The experimental protocols were approved by the Committee on the Ethics of Animal Experiments of Tianjin Medical University General Hospital (IRB2021-KY-161). The mice were randomized allocated to vehicle, 1 h tPA, 5 h tPA and 5 h tPA + HRG group. The mice were anesthetized by inhalation of 1–1.5% isoflurane in a mixture of 30% oxygen and 70% nitrous oxide. Body temperature was maintained at 37 ± 0.5 °C. The skin between the right eye and the right ear was incised, and the temporal muscle was separated and retracted to expose the skull. A projection of the middle cerebral artery on the skull surface was found at the midpoint between the right eye and right ear. The photosensitive dye rose bengal (60 mg/kg; 330000, Sigma-Aldrich, St. Louis, MO, USA) was injected through the tail vein, and 5 min later, the projection area of the middle cerebral artery was illuminated using a 3.5 mW green light laser (540 nm) for 25 min. d-Glucose (6 ml/kg at 50% weight/volume, 158968, Sigma-Aldrich, St. Louis, MO, USA) was injected intraperitoneally 15 min before stroke induction. Human tPA (10 mg/kg; Boehringer Ingelheim, Germany) or saline was administered as an IV bolus injection of 1 mg/kg, followed by a 9 mg/kg continuous infusion for 30 min at 1 h and 5 h after stroke, respectively. HRG (1 mg/ml, volume 3 μl; 10836-H08H, Sino Biological, Beijing, China) or saline was administered by intraventricular injection 10 min before tPA infusion at 5 h after stroke.

## Neurological function assessment

At 24 h after tPA treatment, the Bederson score was used to measure neurological deficits. Rotarod and corner tests were used to assess sensorimotor function among the groups. The researchers who performed neurological function assessment and analysis were kept blind towards the allocation of the mice.

## Quantification of cerebral hemorrhage

At 24 h after stroke, the mice were perfused intracardially with PBS. Hemorrhage in the ischemic hemispheric brain tissue of each mouse was quantified using a hemoglobin content assay kit (BC5585, Solarbio, Beijing, China) according to the manufacturer's protocol.

## Neutrophil depletion and DNase I treatment

Neutrophils were depleted by intraperitoneal (i.p.) injection of 8 mg/kg anti-mouse Ly6G mAb (BE0075-1, Bio X Cell) 24 h before stroke induction. Neutrophil depletion was confirmed via flow cytometry using Ly6G staining (Appendix Fig. S1B). For NETosis inhibition, mice were injected intraperitoneally (i.p.) with 5 mg/kg of DNase I (D5025, Sigma-Aldrich, St. Louis, MO, USA) or saline, immediately after stroke.

## Endogenous HRG depletion by siRNA injection in vitro and in vivo

HRG small interfering RNA (siRNA) was synthesized with the target sequence as follows: 5′-GUUCUAGACC UGAUCAAUA-3′ (Table EV6). To test the efficiency of HRG knockdown in vitro, HRG siRNA or control siRNA was mixed with Lipo3000, and added into the culture system of hepatocytes AML12 cells (CL-0602, Procell Life Science & Technology Co, Ltd, Wuhan, China). The AML12 cell line has been authenticated by Shor Tandem Repeat (STR) DNA genotype analysis, and cell line was not contaminated with mycoplasma. To knockdown the HRG gene in vivo, the Silencer Select predesigned HRG siRNA or siRNA control was combined with Entranster™ in vivo (18668-11-1, Engreen Biosystem Co, Ltd, Beijing, China) for generating complexes, in accordance with the manufacturer's instructions, and the complexes were injected i.p. into C57BL/6 mice at a dose of 4 mg/kg. The knockdown of HRG mRNA in liver was confirmed by qPCR, and the levels of HRG in plasma were validated using ELISA 24 h after injection (Appendix Fig. S4). Stroke was performed 24 h after the injection.

## Immunofluorescence staining

The mice were anesthetized and perfused with PBS, and brain tissue was collected. After fixation with 4% formaldehyde and gradient dehydration with 15% and 30% sucrose solutions, the brains were cut into 8-μm-thick sections. Brain sections were blocked with 3% bovine serum albumin and permeabilized with 0.3% Triton X-100. Rat anti-Ly6G mAb (1:200; 127605, Biolegend, CA, USA) was then added and incubated overnight at 4 °C. Next, the sections were washed with PBS, incubated with Alexa-Fluor 488-conjugated secondary antibody (1:1000; A-21208, Invitrogen, CA, USA) for 1.5 h at room temperature, washed, and mounted. Images were acquired using a fluorescence microscope (Biotek Cytation 5, USA) and analyzed using ImageJ software.

## Real-time quantitative PCR analysis

Total RNA extraction was performed using TRIzol reagent (Invitrogen, USA). Real-time RT-PCR analysis was conducted

utilizing PerfectStart SYBR Green qPCR super mix (AQ601, Transgen, Beijing, China) on the Bio-Rad CFX96 real-time system (Bio-Rad, Hercules, CA) to assess mRNA expression. The comparative threshold cycle $(2-\triangle\triangle CT)$ method was employed for relative mRNA quantification of the target genes. The primer sequences used for RT-PCR were as follows: Glyceraldehyde-3-phosphate dehydrogenase (GAPDH) F-5′ TGG AGA AAC CTG CCA AGT ATGA 3′, R-5′ TGG AAG AAT GGG AGT TGC TGT 3′; HRG F-5′ TGT CTC TTC AGC ACT TCG CA 3′; R-5′ GCC CGT TCC ACT CTG AAA GA 3′ (Table EV6).

## Western blotting analysis

Polyacrylamide gels (12.5%) were used to electrophorese 20 µg of protein extracted from human plasma or platelets, which was then transferred onto a polyvinylidene difluoride (PVDF) membrane (Bio-Rad, Hercules, CA). Following staining with SYPRO Ruby (Life Technologies, USA), the membrane was blocked using 10% skim milk for 1 h and subsequently incubated overnight at 4 °C with rabbit anti-HRG polyclonal antibody (26252-1-AP, Proteintech, Wuhan, China). This was followed by incubation with anti-rabbit IgG goat polyclonal IgG-horseradish peroxidase (HRP) secondary antibody (MBL, Nagoya, Japan) for 2 h at room temperature. Finally, signal visualization was achieved using an enhanced chemiluminescence system (Pierce Biotechnology, Rockford, IL).

## RNA sequencing of neutrophils

Neutrophils cultured in PBS, tPA, or tPA + HRG for 2.5 h were collected. RNA was purified using TRIzol reagent (Thermo Fisher Scientific, Waltham, MA, USA) according to the manufacturer's protocol, and total RNA was used as the input material for RNA sample preparation. Briefly, mRNA was purified from total RNA using poly T oligo-attached magnetic beads, followed by random fragmentation of the obtained mRNA. The first strand of cDNA was synthesized using fragmented mRNA as a template and a random hexamer primer. Subsequently, RNase H was employed to degrade the RNA strand, and the second strand of cDNA was synthesized using dNTPs under DNA polymerase I. To preferentially select cDNA fragments 370–420 bp in length, the library fragments were purified using the AMPure XP system. PCR was conducted using the Phusion High-Fidelity DNA polymerase, Universal PCR primers, and Index (X) Primer. Subsequently, the PCR products were purified using the AMPure XP system. The quality of the library was evaluated on the Agilent Bioanalyzer 2100 system. Clustering of the index-coded samples was carried out on a cBot Cluster Generation System utilizing the TruSeq PE Cluster Kit v3-cBot-HS (Illumina) as per manufacturer's instructions. Following cluster generation, sequencing of the library preparations took place on an Illumina NovaSeq platform to generate 150 bp paired-end reads. Raw data underwent quality control before building an index of the reference genome with Hisat2 v2.0.5 software. Clean paired-end reads were then aligned to this reference genome. The number of reads mapped to each gene was determined using featureCounts v1.5.0-p3 tool. Differential gene expression analysis between two groups utilized DESeq2 R package (version 1.20). Statistical enrichment analysis for differentially expressed genes (DEGs) in KEGG pathways and GO enrichment analysis were performed employing clusterProfiler R package.

### The paper explained

**Problem**

Intravenous thrombolysis with recombinant tissue plasminogen activator (tPA) is the primary treatment for acute ischemic stroke (AIS) within the therapeutic time window of 4.5 h. However, the potential benefits of thrombolysis for stroke are compromised due to the risk of hemorrhagic transformation, particularly with delayed thrombolysis. The pathophysiology of tPA-related hemorrhagic transformation (HT) still needs to be clearly defined, along with its predictive factors and therapeutic strategies.

**Results**

Histidine-rich glycoprotein was found elevated 1 h following thrombolysis, as identified by high-throughput proteomic analysis of plasma. The elevation of HRG was further confirmed in a validation cohort of 157 individuals, which correlated with reduced risk of hemorrhagic transformation. The further experiments in AIS animal model revealed that HRG inhibited neutrophils NETosis, pro-inflammatory responses and migration induced by tPA, which ultimately reduced the hemorrhagic transformation.

**Impact**

Synthesized in the liver, plasma HRG level reflected the risk of thrombolysis-related hemorrhagic transformation. Endogenous HRG serves as a modulator to maintain homeostasis between the fibrinolytic process and immune responses. Monitoring the levels of HRG could be a potential target for predicting and managing thrombolysis-related hemorrhagic transformation.

## Statistical analysis

Sample size was determined empirically based on the previous experience in the calculation of experimental variability. All experiments in this study have been performed independently at least three biological replicates. The Kruskal–Wallis H test was used to compare three or more independent groups of data that were not normally distributed, followed by Dunn's multiple comparisons. The Mann–Whitney U-test was used to compare two independent groups of data that were not normally distributed. Paired t-tests were used to compare the two groups before and after tPA treatment. Outlier testing was conducted on all datasets and any statistical outliers were removed. All data were presented as means ± SEM and analyzed using SPSS 22.0 software. Statistical significance was set at $p < 0.05$.

## For more information

Author's homepage: https://www.ae-info.org/ae/Member/Shi_Fu-Dong.

# Data availability

The transcriptome data for neutrophils under tPA and HRG generated during this current study were deposited to the GEO database (https://www.ncbi.nlm.nih.gov/geo/query/acc.cgi?acc=GSE247434) with the accession number of GSE247434. The mass spectrometry proteomics data have been deposited to the ProteomeXchange Consortium via the

PRIDE partner repository with the dataset identifier PXD046738 and PXD051719.

The source data of this paper are collected in the following database record: biostudies:S-SCDT-10_1038-S44321-024-00117-y.

## Peer review information

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

## Acknowledgements

We acknowledge Dr. Y Liu and Dr. M Li for kindly sharing laboratory equipment and technical expertise during the research. This work was supported by the National Natural Science Foundation of China (82171277, 82201432, 82301523, 81830038) and Tianjin Key Medical Discipline (Specialty) Construction Project.

## Author contributions

**Wei Jiang**: Conceptualization; Data curation; Supervision; Funding acquisition; Writing—original draft; Writing—review and editing. **Yuexin Zhao**: Data curation; Software; Formal analysis. **Rongrong Liu**: Data curation; Validation; Methodology. **Bohao Zhang**: Data curation. **Yuhan Xie**: Resources; Formal analysis. **Bin Gao**: Software; Formal analysis; Visualization. **Kaibin Shi**: Software; Methodology; Project administration. **Ming Zou**: Resources; Software; Formal analysis; Methodology; Project administration. **Dongmei Jia**: Resources; Investigation; Methodology. **Jiayue Ding**: Resources; Funding acquisition; Investigation. **Xiaowei Hu**: Resources. **Yanli Duan**: Resources; Formal analysis; Investigation. **Ranran Han**: Formal analysis; Funding acquisition; Methodology. **DeRen Huang**: Methodology; Writing—original draft; Writing—review and editing. **Luc Van Kaer**: Supervision; Validation; Writing—original draft; Writing—review and editing. **Fu-Dong Shi**: Conceptualization; Resources; Funding acquisition; Writing—original draft; Writing—review and editing.

Source data underlying figure panels in this paper may have individual authorship assigned. Where available, figure panel/source data authorship is listed in the following database record: biostudies:S-SCDT-10_1038-S44321-024-00117-y.

## Disclosure and competing interests statement

LVK is a member of the scientific advisory board of Isu Abxis Co., Ltd. (Republic of Korea). The other authors declare that they have no competing interests.

# Expanded View Figures

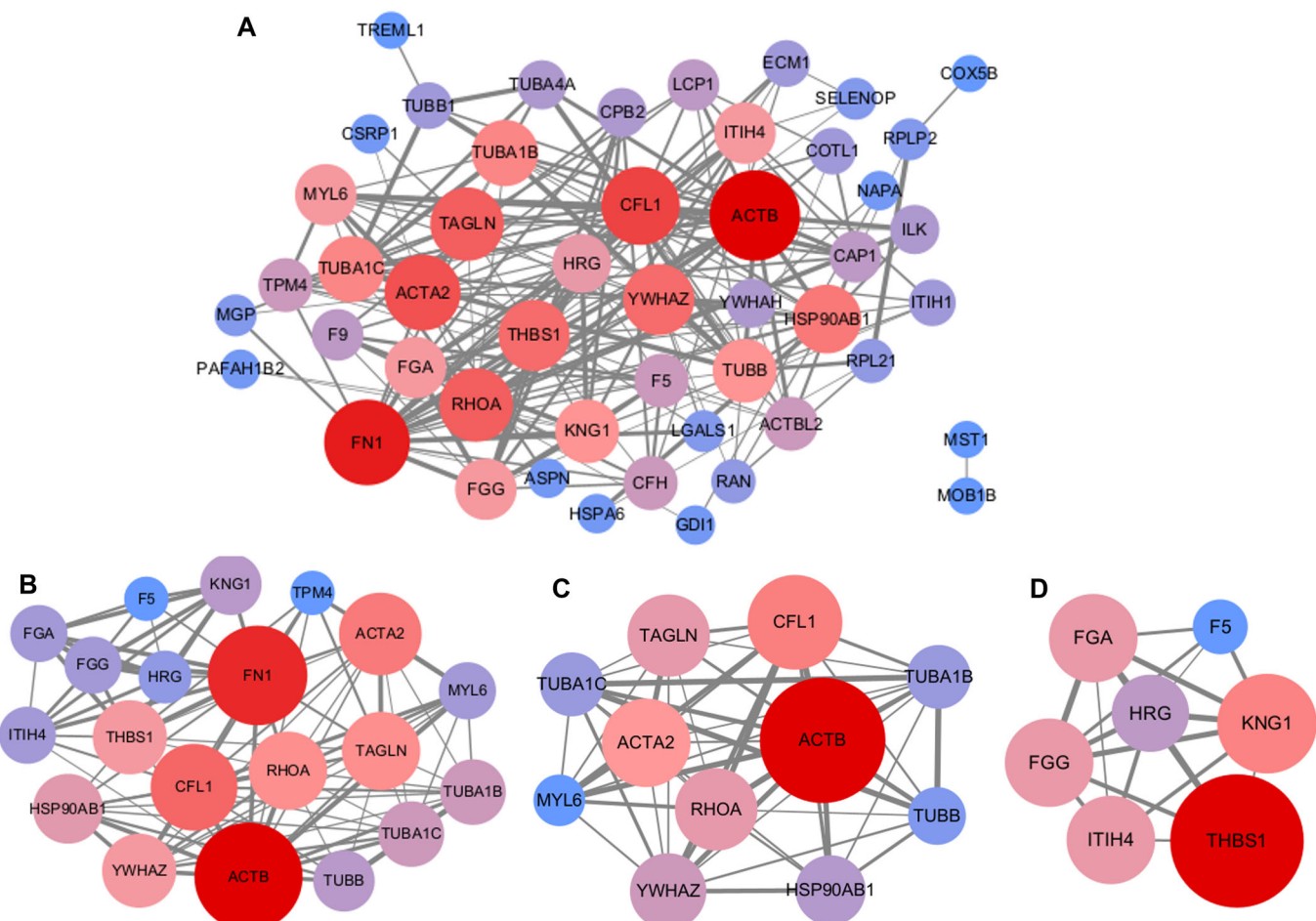

**Figure EV1.   Protein–protein interaction (PPI) network analysis of differentially expressed proteins (DEPs).**

(**A**) PPI network of plasma DEPs between individuals after tPA vs. before and after tPA treatment. The PPI numbers of the DEPs are indicated by node size and color. Edge thickness indicates the degree of interaction between DEPs. (**B**) Top 20 hub genes identified using the CytoHubba plugin. (**C**, **D**) The top two significant cluster networks identified using the MCODE plugin.

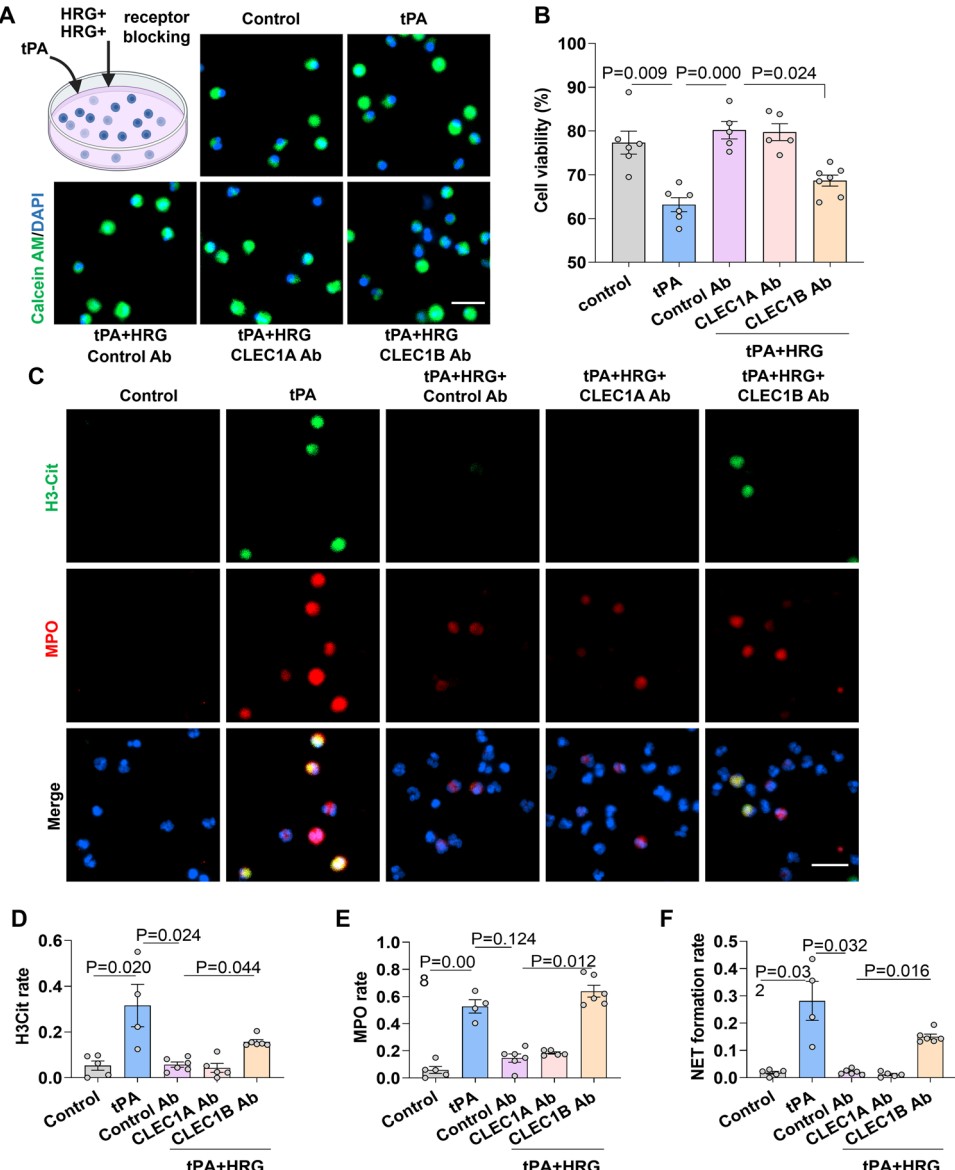

**Figure EV2.   Effect of CLEC1A and CLEC1B blockade on the mortality rate and NETosis of neutrophils in response to tPA and HRG treatment in vitro.**

(**A**) Neutrophils were labeled with calcein-AM (living cell, green) and Hoechst 33342 (nuclei, blue). Neutrophils were cultured under the following conditions: vehicle, tPA, tPA + HRG + Control Ab, tPA + HRG + blocking antibodies against CLEC1A and CLEC1B. After 18 h of culture, the viability of neutrophils was observed and calculated under a fluorescent microscope. Typical results from five independent experiments are shown. Scale bar = 20 μm. (**B**) Statistical analysis of the cell viability is shown (n = 5–7). (**C**) Neutrophils were incubated for 2.5 h and co-stained with H3Cit (green), MPO (red), and DAPI (blue). Merged images indicate NET formation. Scale bar = 20 μm. (**D–F**) Statistical analysis and comparison of H3Cit, MPO, and NET formation rates in different experimental groups (n = 4–6). Data information: Data are presented as mean ± SEM. Two-tailed Kruskal–Wallis H test is used in (**B**, **D–F**). Source data are available online for this figure.

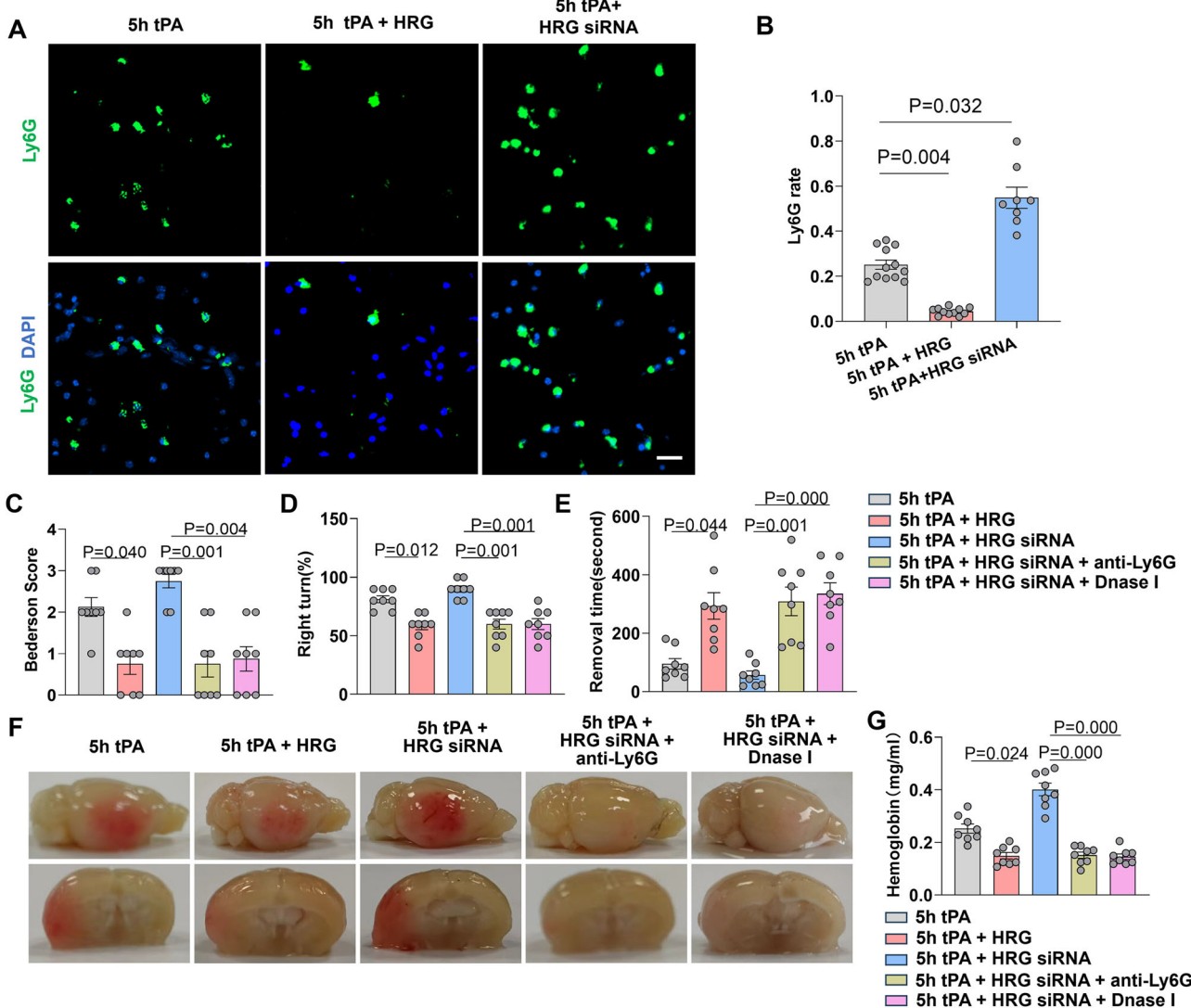

**Figure EV3. Endogenous HRG depletion exacerbates hemorrhagic transformation and neurological deficits, which are rescued by neutrophil depletion or NETosis inhibition.**

(A) Representative images of immunofluorescence staining of infiltrating neutrophils (Ly6G, green) in brain lesions. Scale bar = 20 μm. (B) The statistical analysis of infiltrating neutrophils (Ly6G rate) from the following experimental groups: tPA (5 h) ($n = 12$), tPA (5 h) + HRG ($n = 11$), and tPA (5 h) + HRG siRNA groups ($n = 8$). The data are also partially used in Fig. 6H. (C–E) Neurological deficits were measured by assessing the Bederson score (C), corner test (D) and adhesive removal test (E) in mice treated with tPA (5 h), tPA (5 h) + HRG, tPA (5 h) + HRG siRNA, tPA (5 h) + HRG siRNA + anti-Ly6G, and tPA (5 h) + HRG siRNA + Dnase I ($n = 8$ per group). (F) Representative images of the dorsal surface and a coronal section showing cerebral hemorrhage 24 h after stroke. (G) Quantification of cerebral hemorrhage by spectrophotometric hemoglobin assay ($n = 8$). Data information: Data are presented as mean ± SEM. Two-tailed Kruskal–Wallis H test is used in (B–E, G). Source data are available online for this figure.

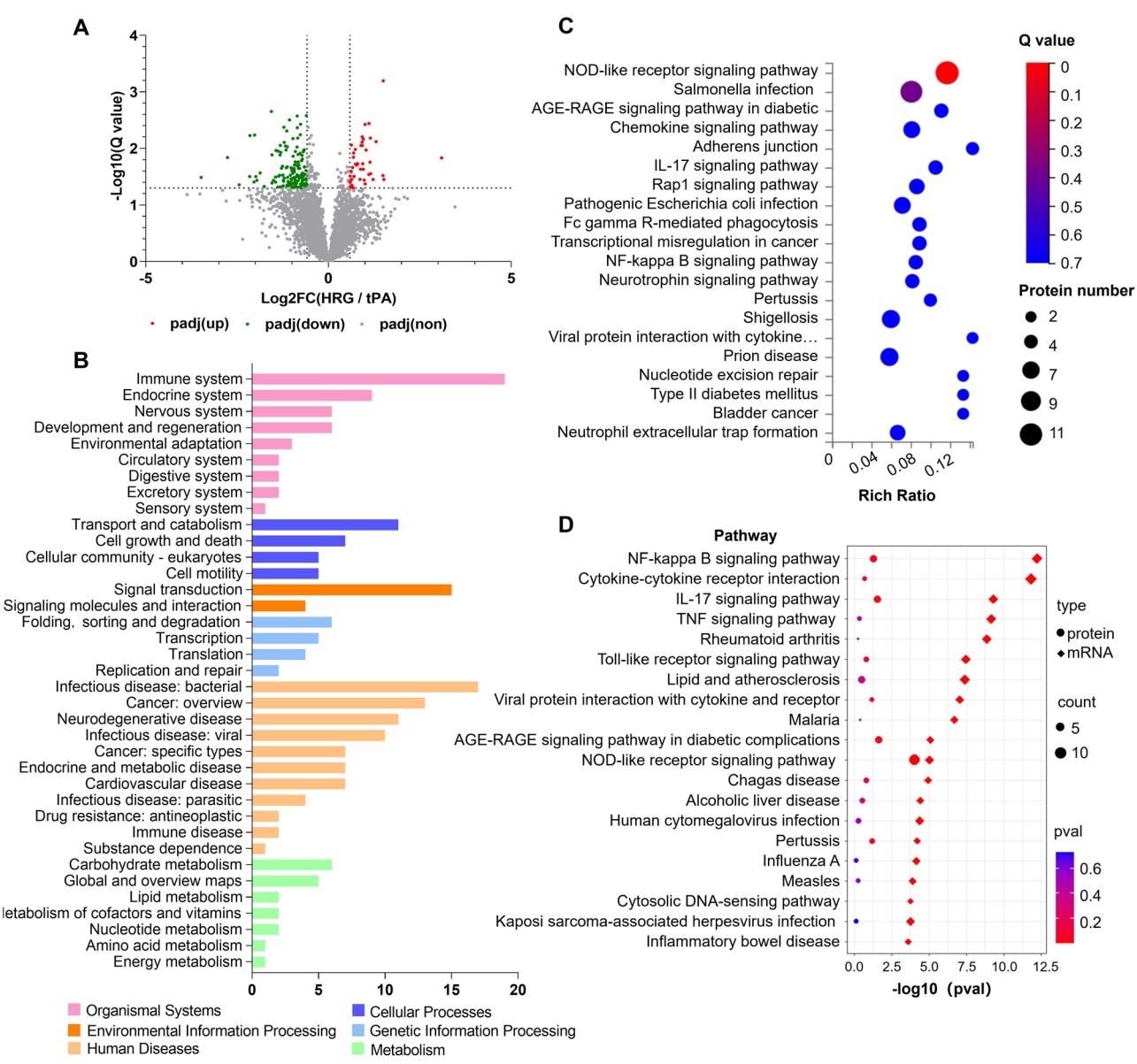

**Figure EV4. Proteomic analysis and correlated differentially expressed genes and proteins of neutrophilia under HRG treatment.**

(**A**) The protein profiles of neutrophils under tPA + HRG vs. tPA treatments were analyzed by the data-independent acquisition (DIA) mode ($n = 6$ per group). DEPs between two groups were identified as the padj <0.05 and fold change >1.5. The volcano plots are shown. (**B**) The downregulated DEPs were analyzed by KEGG classification in neutrophils with tPA + HRG vs. tPA treatment groups. (**C**) Bubble plot of KEGG pathway analysis of downregulated DEPs. Rich ratio represents the ratio of the number of target proteins vs. all proteins included in each pathway. Dot size represents the relative number of enriched proteins. The color of the dot represents the size of the Q-value (padj). (**D**) KEGG pathway enrichment analysis of correlated (Cor)-DEGs-DEPs. The enrichment of the proteomics is represented by circles, while the enrichment of the transcriptomics is represented by diamonds. Dot size represents the relative number of enriched proteins. The color of the dot represents the size of the Q-value (padj). For statistical methods, we performed hypergeometric test and Benjamini-Hochberg correction for Q-value (padj). We performed KEGG functional enrichment for DEGs and DEPs by clusterProfiler R package (3.8.1).

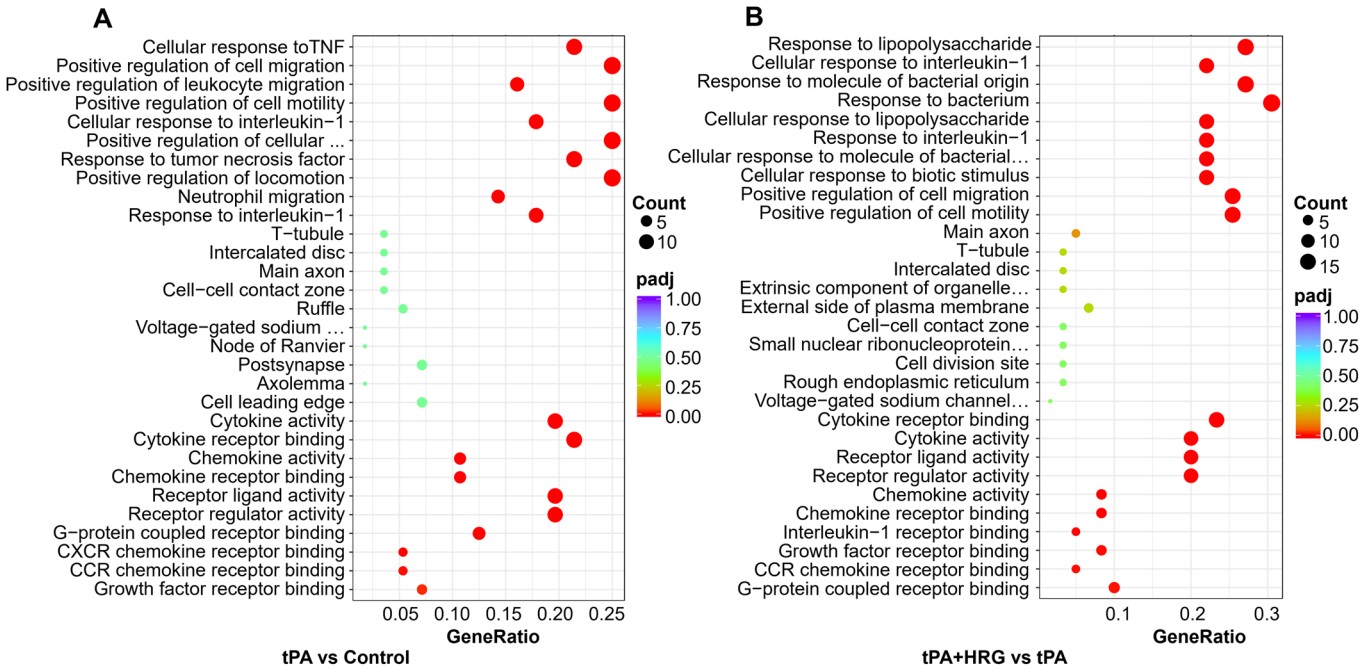

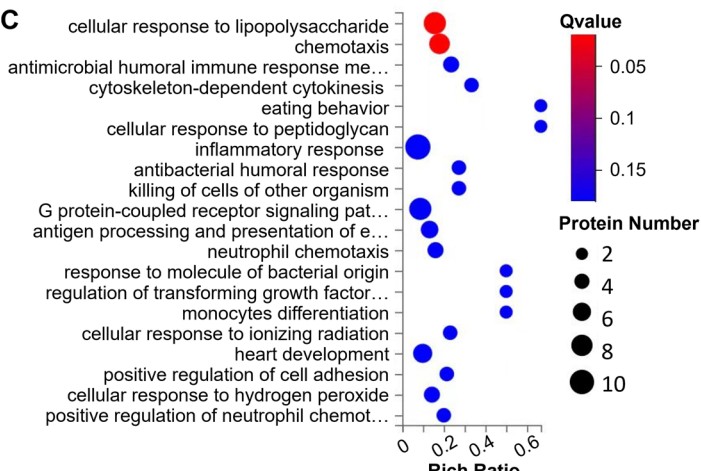

**Figure EV5. Gene Ontology (GO) analysis following RNA and proteomic sequencing of purified neutrophils with recombinant tissue plasminogen activator (tPA) and histidine-rich glycoprotein (HRG) treatment.**

(**A**) Functional GO enrichment analysis of the tPA vs. control group based on RNA sequencing. Enriched biological processes included cellular responses to inflammation, cell migration, and leukocyte migration. The most enriched molecular functions were cytokine activity, cytokine receptor binding, and chemokine activity. There was no significant enrichment in the cell components. We performed GO enrichment for DEGs by clusterProfiler R package (3.8.1) and we used hypergeometric test and Benjamini-Hochberg (BH) correction for padj. (**B**) Functional GO enrichment for the tPA + HRG vs. the tPA group based on RNA sequencing. The enriched biological processes mainly included the response to infectious stimulation, cellular response to IL-1, cell migration, and cell motility. Cytokine receptor binding, cytokine activity, and receptor ligand activity were significantly enriched in molecular functions. There was no significant enrichment in the cell components. We performed GO enrichment for DEGs by clusterProfiler R package (3.8.1) and we used hypergeometric test and BH correction for padj. (**C**) Functional GO enrichment for the tPA + HRG vs. the tPA group based on proteomic sequencing. The enriched biological processes mainly included the response to infectious stimulation and chemotaxis. The annotation function of the proteins was analyzed by GO using the online DAVID software, and we used hypergeometric test and BH correction for Q-value (padj).

