## [Peer Review File · EMBO Molecular Medicine]

Histidine-rich glycoprotein modulates neutrophils and thrombolysis-associated hemorrhagic transformation

Wei Jiang, Yuexin Zhao, Rongrong Liu, Bohao Zhang, Yuhan Xie, Bin Gao, Kaibin Shi, Ming Zou, Dongmei Jia, Jiayue Ding, Xiaowei Hu, Yanli Duan, Ranran Han, Deren Huang, Luc Van Kaer, and Fu-Dong Shi

Corresponding author: Fu-Dong Shi (fshi@tmu.edu.cn)

Review Timeline:

Submission Date:	7th Jan 24
Editorial Decision:	5th Feb 24
Revision Received:	24th May 24
Editorial Decision:	5th Jul 24
Revision Received:	18th Jul 24
Accepted:	29th Jul 24

Editor: Poonam Bheda

Transaction Report:

5th Feb 2024

Dear Prof. Shi,

Thank you for the submission of your manuscript to EMBO Molecular Medicine. We have now received feedback from the two reviewers who agreed to evaluate your manuscript. As you will see from the reports below, the referees acknowledge the interest of the study and are overall supportive of your work; however they also comment on multiple aspects of the manuscript that should be strengthened in a revision. Without reiterating all of their comments, in particular Reviewer 1 has emphasized mechanisms and causal effects should be improved, Reviewer 2 has technical concerns on the ex vivo neutrophil integrity and suggests HRG depletion should be tested, and Reviewer 3 has concerns on the choice of statistical tests. Given the translational nature of EMBO Molecular Medicine, editorially we focus on the clinical and therapeutic aspects of a manuscript, and therefore while we agree that additional insight into mechanisms would be beneficial to the study, providing some additional support without a full mechanistic understanding will be sufficient for publication in EMM.

Addressing the reviewers' concerns in full in a point-by-point response will be necessary for further considering the manuscript in our journal, and acceptance of the manuscript will entail a second round of review. EMBO Molecular Medicine encourages a single round of revision only and therefore, acceptance or rejection of the manuscript will depend on the completeness of your responses included in the next, final version of the manuscript. For this reason, and to save you from any frustrations in the end, I would strongly advise against returning an incomplete revision. If you would like to discuss further the points raised by the referees, I am available to do so via email or video. Let me know if you are interested in this option.

We are expecting your revised manuscript within three months, if you anticipate any delay, please contact us. When submitting your revised manuscript, please carefully review the instructions that follow below. We perform an initial quality control of all revised manuscripts before re-review; failure to include requested items will delay the evaluation of your revision.

We require:

- 1) A .docx formatted version of the manuscript text (including legends for main figures, EV figures and tables). Please make sure that the changes are highlighted to be clearly visible.
- 2) Individual production quality figure files as .eps, .tif, .jpg (one file per figure). For guidance, download the 'Figure Guide PDF' (<https://www.embopress.org/page/journal/17574684/authorguide#figureformat>).
- 3) At EMBO Press we ask authors to provide source data for the main figures. Our source data coordinator will contact you to discuss which figure panels we would need source data for and will also provide you with helpful tips on how to upload and organize the files.
- 4) A .docx formatted letter INCLUDING the reviewers' reports and your detailed point-by-point responses to their comments. As part of the EMBO Press transparent editorial process, the point-by-point response is part of the Review Process File (RPF), which will be published alongside your paper.
- 5) A complete author checklist, which you can download from our author guidelines (<https://www.embopress.org/page/journal/17574684/authorguide#submissionofrevisions>). Please insert information in the checklist that is also reflected in the manuscript. The completed author checklist will also be part of the RPF.
- 6) Please note that all corresponding authors are required to supply an ORCID ID for their name upon submission of a revised manuscript.
- 7) It is mandatory to include a 'Data Availability' section after the Materials and Methods. Before submitting your revision, primary datasets produced in this study need to be deposited in an appropriate public database, and the accession numbers and database listed under 'Data Availability'. Please remember to provide a reviewer password if the datasets are not yet public (see <https://www.embopress.org/page/journal/17574684/authorguide#dataavailability>).

This study includes no data deposited in external repositories.

- 8) For data quantification: please specify the name of the statistical test used to generate error bars and P values, the number (n) of independent experiments (specify technical or biological replicates) underlying each data point and the test used to calculate p-values in each figure legend. The figure legends should contain a basic description of n, P and the test applied. Graphs must include a description of the bars and the error bars (s.d., s.e.m.). Please provide exact p values.

10) We replaced Supplementary Information with Expanded View (EV) Figures and Tables that are collapsible/expandable online. A maximum of 5 EV Figures can be typeset. EV Figures should be cited as "Figure EV1, Figure EV2" etc... in the text and their respective legends should be included in the main text after the legends of regular figures.

13) Author contributions: CRedit has replaced the traditional author contributions section because it offers a systematic machine readable author contributions format that allows for more effective research assessment. Please remove the Authors Contributions from the manuscript and use the free text boxes beneath each contributing author's name in our system to add specific details on the author's contribution. More information is available in our guide to authors.

Please also suggest a striking image or visual abstract to illustrate your article as a PNG file 550 px wide x 300-600 px high. Share synopsis text and image, as well as eTOC:

Please note that these would be the final versions and changes during proofing are usually not allowed

16) As part of the EMBO Publications transparent editorial process initiative (see our policy here:

https://www.embopress.org/transparent-process#Review_Process), EMBO Molecular Medicine will publish online a Peer Review File (PRF) to accompany accepted manuscripts.

In the event of acceptance, this file will be published in conjunction with your paper and will include the anonymous referee reports, your point-by-point response and all pertinent correspondence relating to the manuscript. Let us know whether you agree with the publication of the PRF and as here, if you want to remove or not any figures from it prior to publication.

EMBO Molecular Medicine has a "scooping protection" policy, whereby similar findings that are published by others during review or revision are not a criterion for rejection. Should you decide to submit a revised version, I do ask that you get in touch

after three months if you have not completed it, to update us on the status.

I look forward to receiving your revised manuscript.

Yours sincerely,

Poonam Bheda

Poonam Bheda, PhD
Scientific Editor
EMBO Molecular Medicine

**** Reviewer's comments ****

Referee #1 (Remarks for Author):

This manuscript explores the impact of Histidine-rich glycoprotein (HRG) in patients with acute ischemic stroke treated with tissue plasminogen activator (tPA). The study suggests a potential protective role of HRG against hemorrhagic transformation by influencing neutrophil mortality and inflammatory responses. The study stands out for its thorough combination of clinical exploration alongside in vitro and in vivo experiments. However, it falls short in clearly defining the mechanism behind the proposed phenotype and in establishing a causal relationship between tPA effects on neutrophils, suggesting that the findings might be circumstantial.

Major specific remarks:

1. The manuscript presents a solid clinical biomarker study, commendable for its use of both an exploratory and a larger validation cohort. The data convincingly identifies HRG as a biomarker in stroke patients, but it would benefit from a comparison with previous stroke biomarker studies or data from larger consortia, such as the UK Biobank, to assess if HRG has been indicated in other cohorts.
2. A significant gap is the unclear cellular or molecular source of HRG. Experimental clarification is needed. Furthermore, the mechanism by which HRG influences neutrophil NETosis is intriguing but not well-explained. The specific receptors on neutrophils engaged by HRG should be identified.
3. Another critical point is the necessity of HRG's effect on neutrophils in mediating hemorrhagic transformation post-stroke. The manuscript does not establish this causal chain, which is vital to its hypothesis. Investigating whether preventing NETosis through neutrophil depletion or pharmacological inhibitors can avert tPA-induced hemorrhagic transformation could be enlightening.
4. The interpretation of transcriptomic data in Figure 7 is ambiguous. It's unclear if the potential effects mediated by tPA or HRG are transcriptionally regulated. The immediacy of HRG's impact on tPA-mediated NETosis suggests a non-transcriptional mechanism, necessitating better integration of this information with proteomic studies and cellular analysis results.

Referee #2 (Comments on Novelty/Model System for Author):

The observation periods for the survival of human neutrophils were not appropriate.

Referee #2 (Remarks for Author):

In this manuscript, the authors tried to demonstrate the t-PA treatment-related changes in plasma proteomics by LC-MS/MS analysis in acute ischemic stroke (AIS) patients. The authors identified histidine-rich glycoprotein (HRG) as one of up-regulated plasma factors. Although the plasma levels of HRG did not change in AIS patients without t-PA treatment, the authors found that the plasma levels of HRG were increased in patients treated with t-PA. Starting from these observations, the authors examined the effects of HRG on t-PA-induced death of purified human neutrophils, NETosis and migration. Furthermore, the authors examined the effects of HRG on mouse model of AIS treated with t-PA. The experimental design seems good, however, I have several concerns on the manuscript listed below.

1. Page 10, line 17:

The concentration of HRG administered to MCAO mice was expressed as 1 microM. Then, how much amount of HRG (dose) did the authors administer to mice ?

2. It is well known that the life span of neutrophils is very short once neutrophils are purified from peripheral blood in a plasma-free condition.

In the migration assay, the authors pretreated neutrophils for 2h under different condition and applied these cells on the monolayer of endothelial cells to observe the migration up to 8 h. This protocol must be harsh enough for the survival of neutrophils. For example, 4 hour count of tPA+ HRG was lower than that of 2 hour count. In Fig.3 experiments, the authors examined cell death 18 hours after the start of incubation. These conditions appear not to be tolerable for neutrophils.

3. In the literatures, siRNA for HRG has been used to reduce the expression of HRG in the liver and to obtain the acute and efficient lowering of plasma HRG in in vivo experiments. The authors should examine the effects of depletion of endogenous HRG on hemorrhagic transformation in the present model.

4. HRG is produced mainly in the liver. Therefore, the authors should provide data on the expression of HRG mRNA after t-PA treatment.

5. Minor point, page 9, line 17:
3 mm pores ? Is this correct ?

Referee #3 (Comments on Novelty/Model System for Author):

1- Authors should add in the abstract the meaning of HRG the first time they cite it.

2- The introduction contains the conclusions of the study. I think the introduction should explain the current state of the art and the reasons leading the authors to investigate but not the results or the conclusions of the study.

3- Methodology: in the abstract it is stated that the study was validated in a cohort of 97 patients but then from methodology it results that the three groups considered were quite unbalanced, with 19 healthy volunteers as control, 61 patients receiving tPA and 17 patients without tPA. This could affect the results of the study, so the authors should include similar number of patients for the several groups of the study or perform some statistical analysis considering this bias and include this as limitation.

4- Authors should add tables with the values of HRG in the control and in the 2 patient groups with all the relevant stats

5- Kruskal-Wallis test should be used rather than Mann-Whitney U-test as I get you have 3 groups to compare: control, tPA yes and tPA no.

Referee #3 (Remarks for Author):

1- Authors should add in the abstract the meaning of HRG the first time they cite it.

2- The introduction contains the conclusions of the study. I think the introduction should explain the current state of the art and the reasons leading the authors to investigate but not the results or the conclusions of the study.

3- Methodology: in the abstract it is stated that the study was validated in a cohort of 97 patients but then from methodology it results that the three groups considered were quite unbalanced, with 19 healthy volunteers as control, 61 patients receiving tPA and 17 patients without tPA. This could affect the results of the study, so the authors should include similar number of patients for the several groups of the study or perform some statistical analysis considering this bias and include this as limitation.

4- Authors should add tables with the values of HRG in the control and in the 2 patient groups with all the relevant stats

5- Kruskal-Wallis test should be used rather than Mann-Whitney U-test as I get you have 3 groups to compare: control, tPA yes and tPA no.

Referee #1 :

This manuscript explores the impact of Histidine-rich glycoprotein (HRG) in patients with acute ischemic stroke treated with tissue plasminogen activator (tPA). The study suggests a potential protective role of HRG against hemorrhagic transformation by influencing neutrophil mortality and inflammatory responses. The study stands out for its thorough combination of clinical exploration alongside in vitro and in vivo experiments. However, it falls short in clearly defining the mechanism behind the proposed phenotype and in establishing a causal relationship between tPA effects on neutrophils, suggesting that the findings might be circumstantial.

Major specific remarks:

1. The manuscript presents a solid clinical biomarker study, commendable for its use of both an exploratory and a larger validation cohort. The data convincingly identifies HRG as a biomarker in stroke patients, but it would benefit from a comparison with previous stroke biomarker studies or data from larger consortia, such as the UK Biobank, to assess if HRG has been indicated in other cohorts.

Reply: Thank you for the thorough review of our manuscript and for your valuable feedback. There are various studies investigating biomarkers of hemorrhagic transformation (HT) in ischemic stroke. However, differences in the timing of blood sampling in these studies limits performing extensive comparisons across studies and our results. Studies with only pre-tPA samples revealed that higher levels of matrix metalloproteinase-9 (MMP-9) (Montaner, Molina et al., 2003), serum cellular-fibronectin (c-Fn) (Castellanos, Leira et al., 2004) and S100B (Foerch, Wunderlich et al., 2007) at baseline are associated with HT in patients treated with tPA (Castellanos, Sobrino et al., 2007). The biomarker studies from pre- vs. post-tPA treatment comparisons were mostly collected at baseline and at 24h post-tPA treatment, among which MMP-9 (Inzitari, Giusti et al., 2013), C-reactive protein (CRP) (Karlinski, Bembenek et al., 2014) and IL-10 (Gori, Giusti et al., 2017) were reported related to HT prediction. Only one study compared samples collected at baseline and within 3h post-tPA treatment, which showed the presence of the HRG protein precursor among degraded protein

fragments post-tPA administration (Ning, Sarracino et al., 2010). Ning et al reported differences in protein expression before and 3 hours after thrombolysis in patients with stroke. They demonstrated that t-PA treatment changed the plasma degradomic profiles in patients with acute stroke, and HRG precursor was found among the degraded protein fragments in degradomic fractions post-tPA administration. However, no information on quantitative protein variations were reported due to limitations in proteomic quantification (Ning et al., 2010). The descriptions above have now been added to the discussion section (Page 15, Line 20).

The UK biobank has included large-scale biomedical data, which contains genetic information, health information and analyses of biological samples. However, no information on samples prior to and 1 hour after thrombolysis in acute ischemic stroke was provided, and these data can therefore not be compared with the proteomic information from our study.

2. A significant gap is the unclear cellular or molecular source of HRG. Experimental clarification is needed. Furthermore, the mechanism by which HRG influences neutrophil NETosis is intriguing but not well-explained. The specific receptors on neutrophils engaged by HRG should be identified.

Reply: We appreciate the constructive feedback. The origin of HRG was further determined and quantified following tPA treatment in photothrombotic middle cerebral artery occlusion (MCAO) mice and human. HRG is synthesized by liver parenchymal cells and is constitutively present in plasma at a concentration of 60–150 mg/ml (0.8–2.0 mM) (Koide, Foster et al., 1986, Poon, Patel et al., 2011, Saito, Goodnough et al., 1982). HRG has been detected on the cell surface of macrophages and monocytes (Sia, Rylatt et al., 1982) and in the granules of platelets and megakaryocytes (Leung, Harpel et al., 1983). The increase of HRG was also found in MCAO mice treated with tPA, starting as early as 1 h post-tPA treatment (**Appendix Fig S2**). To further clarify the origin of plasma HRG following tPA treatment, cells from liver, peripheral blood and various tissues of mice after MCAO and tPA treatment were collected for HRG determination. HRG mRNA expression in the liver increased as early as 1h post-tPA treatment, and slightly decreased at 4h post-tPA treatment (**Fig. 1A for reviewers**). However, HRG mRNA could not be detected in immune tissues including the spleen, thymus, lymph node, bone marrow and peripheral blood leucocytes in C57BL/6 mice following tPA treatment, which is consistent with a previous report (Hulett & Parish, 2000) (**Fig. 1B for**

reviewers). In addition, it has been shown that HRG is present in platelets and released upon thrombin stimulation in human blood (Leung et al., 1983). To determine if platelet-derived HRG was a significant cellular source of plasma HRG, platelet-rich plasmas (PRP) was collected from human peripheral blood and stimulated with tPA in vitro. After 40min of incubation, PRP was centrifuged for platelet-poor plasma (PPP) and platelet separation. Proteins were extracted from PRP, PPP and platelets for Western Blot analysis (**Fig. 1C for reviewers**). No increase in HRG production was detected in PRP, PPP or platelets after tPA stimulation (**Fig. 1D for reviewers**). Therefore, it indicated that HRG primarily synthesized in the liver contributed to HRG elevation in plasma after tPA stimulation.

Figure 1 for Reviewers (also shown as Appendix Figure S6 in revised manuscript). HRG expression in various tissues following tPA treatment. (A) HRG mRNA expression in the liver after MCAO induction and tPA administration (n=5-7). (B) HRG mRNA expression in the spleen, thymus, lymph node, bone marrow and peripheral blood leukocytes in MCAO mice following tPA treatment (n=3). (C) Platelet-rich plasmas (PRP) was collected from human peripheral blood and stimulated with tPA in vitro. Then PRP was centrifuged for platelet-poor plasma (PPP) and platelet separation. HRG protein (70KD) was detected by Western blot analysis. (D) Statistical analysis of relative HRG protein expression (n = 4). Data information: Results are shown as mean ± SEM. Two tailed Kruskal-Wallis H test is used in (A, D).

HRG has been reported to bind to various components and receptors, which include the C-type lectin receptors CLEC1A (Gao, Wake et al., 2020, Takahashi, Wake et al., 2021) and CLEC1B (Nishimura, Wake et al., 2019). To identify the specific receptors on neutrophils engaged by HRG, blocking antibodies against CLEC1A and CLEC1B were added to the tPA-treated neutrophils during culture, followed by HRG treatment (**Fig. 2 for reviewers**). The decreased mortality rate induced by HRG was reversed in the anti-CLEC1B Ab treatment group, indicating that CLEC1B might be involved in mediating the protective role of HRG towards tPA-stimulated neutrophils (**Fig. 2B for reviewers**). Furthermore, NET formation was evaluated in the anti-CLEC1A and anti-CLEC1B Abs treatment groups (**Fig. 2C for reviewers**). We found that the decreased NET formation in response to HRG treatment was reversed in the anti-CLEC1B Ab treatment group, but not the anti-CLEC1A Ab treatment group (**Fig. 2D-F for reviewers**). Therefore, the above results indicated that CLEC1B might play an essential role in mediating the protective effect of HRG towards neutrophil NETosis.

Figure 2 for Reviewers (also shown as Figure EV.2 in revised manuscript). Effect of CLEC1A and CLEC1B blockade on the mortality rate and NETosis of neutrophils in response to tPA and HRG treatment in vitro. (A) Neutrophils were labeled with calcein-AM (living cell, green) and Hoechst 33342 (nuclei, blue). Neutrophils were cultured under the following conditions: vehicle, tPA, tPA + HRG + Control Ab, tPA + HRG + blocking antibodies against CLEC1A and CLEC1B. After 18 h of culture, the viability of neutrophils was observed and calculated under a fluorescent microscope. Typical results from five independent experiments are shown. Scale bar = 20 μ m. (B) Statistical analysis of the cell viability is shown (n = 5–7). (C) Neutrophils were incubated for 2.5 h and co-stained with H3Cit (green), MPO (red), and DAPI (blue). Merged images indicate NET formation. Scale bar = 20 μ m. (D-F) Statistical analysis and comparison of H3Cit, MPO, and NET formation rates in different experimental groups (n = 4–6). Data information: Data are presented as

mean \pm SEM. Two tailed Kruskal-Wallis H test is used in (B, D-F).

3. Another critical point is the necessity of HRG's effect on neutrophils in mediating hemorrhagic transformation post-stroke. The manuscript does not establish this causal chain, which is vital to its hypothesis. Investigating whether preventing NETosis through neutrophil depletion or pharmacological inhibitors can avert tPA-induced hemorrhagic transformation could be enlightening.

Reply: We appreciate the valuable feedback and valid concerns. HRG administration was found protective towards delayed-thrombolysis-induced HT and neutrophil infiltration. To shed light on the necessity of HRG's effect, we employed siRNA to deplete endogenous HRG and investigated its effect on HT. We showed that siRNA for HRG was effective in reducing HRG mRNA expression in the hepatocytes cell line AML12 in vitro (**Fig.3A for reviewers**). Additionally, we showed that siRNA treatment reduced HRG mRNA expression in the liver (**Fig.3B for reviewers**) and rapidly reduced plasma HRG levels in vivo (**Fig.3C for reviewers**). Next, we treated mice with siRNA, induced MCAO, and performed delayed-tPA treatment. We found that neutrophil infiltration, HT and neurological deficits were exacerbated in the siRNA treatment group. Importantly, neutrophil depletion (anti-Ly6G) or NETosis inhibitor (DNase I) treatment rescued the protective effects of HRG in vivo (**Fig.4 for Reviewers**). These findings suggested that the protective effect of HRG on HT was mediated by inhibiting neutrophils and NETosis.

Figure 3 for Reviewers (also shown as Appendix Figure S5 in revised manuscript). HRG expression decreases following siRNA delivery in vitro and in vivo. (A) HRG siRNA was delivered to AML12 hepatocytes in culture, which showed decreased mRNA expression (n=6). **(B)** HRG siRNA was injected into C57BL/6 mice at a dose of 1 mg/kg via the tail vein. HRG mRNA expression in liver was determined 24 hours after the injection, which showed decreased mRNA levels (17 \pm 0.07%) compared to the control siRNA group (n=5). **(C)** HRG levels in plasma were determined by ELISA, which showed significant decreases following HRG siRNA treatment (n=6).

Data information: Results are shown as mean \pm SEM. Two tailed Mann-Whitney U-test is used in (A-C).

Figure 4 for Reviewers (also shown as Figure EV.3 in revised manuscript). Endogenous HRG depletion exacerbates hemorrhagic transformation and neurological deficits, which are rescued by neutrophil depletion or NETosis inhibition. (A) Representative images of immunofluorescence staining of infiltrating neutrophils (Ly6G, green) in brain lesions. (B) The statistical analysis of infiltrating neutrophils (Ly6G rate) from the following experimental groups: tPA (5 h) (n = 12), tPA (5 h) + HRG (n = 11), and tPA (5 h) + HRG siRNA groups (n = 8). The data are also partially used in Fig 6H. (C-E) Neurological deficits were measured by assessing the Bederson score (C), corner test (D) and adhesive removal test (E) in mice treated with tPA (5 h), tPA (5 h) + HRG, tPA (5 h) + HRG siRNA, tPA (5 h) + HRG siRNA + anti-Ly6G, and tPA (5 h) + HRG siRNA + Dnase I (n = 8 per group). (F) Representative images of the dorsal surface and a coronal section showing cerebral hemorrhage 24 h after stroke. (G) Quantification of cerebral hemorrhage by spectrophotometric hemoglobin assay (n = 8). Data information: Data are presented as mean \pm SEM. Two tailed Kruskal-Wallis H test is used in (B-E, G).

4. *The interpretation of transcriptomic data in Figure 7 is ambiguous. It's unclear if the potential effects mediated by tPA or HRG are transcriptionally regulated. The immediacy of HRG's impact on tPA-mediated NETosis suggests a non-transcriptional mechanism, necessitating better integration of this information with proteomic studies and cellular analysis results.*

Reply: Thank you for the valuable feedback, based on which we performed proteomic analyses on neutrophils treated with or without HRG. The protein profile of neutrophils was determined through high-throughput MS-based proteomics using a data-independent acquisition (DIA) strategy.

Differentially expressed proteins (DEPs) between the two groups were identified as the $\text{padj} < 0.05$ and fold change > 1.5 . A total of 183 DEPs in the tPA + HRG vs. tPA groups were identified, which included 43 upregulated and 140 downregulated proteins, as shown in **Fig. 5A for reviewers**. The down-regulated DEPs under HRG treatment were analyzed by KEGG classification, which revealed enrichment for immune system pathways (**Fig. 5B for reviewers**). The KEGG pathway analysis indicated that the down-regulated DEPs were enriched in pro-inflammatory signaling pathways, which include the NOD (nucleotide-binding oligomerization domain)-like receptor signaling pathway, the interleukin-17 (IL-17) signaling pathway, and the transcription factor nuclear factor kappa B (NF- κ B) signaling pathway (**Fig. 5C for reviewers**). An integrated analysis of transcriptomic and proteomic data provided a comprehensive overview for the down-regulation of neutrophil signaling pathways following HRG treatment. The combined analysis indicated that the correlated (Cor)-DEGs-DEPs were significantly enriched in the NOD-like receptor signaling pathway, at both mRNA and protein levels (**Fig. 5D for reviewers**). The NOD-like receptor is responsible for detecting molecular patterns signals in the cytosol and initiates a signal cascade that eventually leads to NF- κ B activation and production of pro-inflammatory cytokines {Inohara, 2001 #19} {Shaw, 2008 #20}. Gene ontology (GO) analysis of the proteomic data also supported the down-regulation of chemotaxis and immune responses pathways following HRG treatment (**Figure EV. 5C**). Therefore, the proteomic and transcriptomic data indicated that HRG inhibited neutrophil pro-inflammatory responses, likely by down-regulating the NOD-like receptor signaling and the downstream NF- κ B activation.

Figure 5 for reviewers (also shown as Figure EV.4 in revised manuscript). Proteomic analysis and correlated differentially expressed genes and proteins of neutrophilia under HRG treatment. (A) The protein profiles of neutrophils under tPA + HRG vs. tPA treatments were analyzed by the data independent acquisition (DIA) mode (n = 6 per group). DEPs between two groups were identified as the padj < 0.05 and fold change > 1.5. The volcano plots are shown. **(B)** The down-regulated DEPs were analyzed by KEGG classification in neutrophils with tPA + HRG vs. tPA treatment groups. **(C)** Bubble plot of KEGG pathway analysis of down-regulated DEPs. Rich ratio represents the ratio of the number of target proteins vs. all proteins included in each pathway. Dot size represents the relative number of enriched proteins. The color of the dot represents the size of the Q-value (padj). **(D)** KEGG pathway enrichment analysis of correlated (Cor)-DEGs-DEPs. The enrichment of the proteomics is represented by circles, while the enrichment of the transcriptomics is represented by diamonds. Dot size represents the relative number of enriched proteins. The color of the dot represents the size of the Q-value (padj).

Referee #2 :

In this manuscript, the authors tried to demonstrate the t-PA treatment-related changes in plasma proteomics by LC-MS/MS analysis in acute ischemic stroke (AIS) patients. The authors identified histidine-rich glycoprotein (HRG) as one of up-regulated plasma factors. Although the plasma levels of HRG did not change in AIS patients without t-PA treatment, the authors found that the plasma levels of HRG were increased in patients treated with t-PA. Starting from these observations, the authors examined the effects of HRG on t-PA-induced death of purified human neutrophils, NETosis and migration. Furthermore, the authors examined the effects of HRG on mouse model of AIS treated with t-PA. The experimental design seems good, however, I have several concerns on the manuscript listed below.

1. Page 10, line 17: The concentration of HRG administered to MCAO mice was expressed as 1 microM. Then, how much amount of HRG (dose) did the authors administer to mice?

Reply: We apologize for the oversight and have corrected this issue in the revised manuscript. HRG (1mg/ml, volume 3µl) was administered by intraventricular injection to MCAO mice before tPA infusion. The total amount of HRG delivered was 3µg per mouse. Therefore, the final concentration of HRG in the cerebral ventricle was approximately 1 µM, assuming the volume of the ventricular system is around 40 µl {Simon, 2016 #21}. The revised description has been added in the revised manuscript (Page 24, Line 5).

2. It is well known that the life span of neutrophils is very short once neutrophils are purified from peripheral blood in a plasma-free condition. In the migration assay, the authors pretreated neutrophils for 2h under different condition and applied these cells on the monolayer of endothelial cells to observe the migration up to 8 h. This protocol must be harsh enough for the survival of neutrophils. For example, 4 hours count of tPA+ HRG was lower than that of 2 hour count. In Fig.3 experiments, the authors examined cell death 18 hours after the start of incubation. These conditions appear not to be tolerable for neutrophils.

Reply: Thank you for the insightful questions raised regarding our study. Neutrophils have an extremely short lifespan in the peripheral circulation, initiating death relatively shortly after their release from the bone marrow. Following apoptosis initiation, neutrophils functions decline, which impacts the accuracy of experimental results. However, measurements of neutrophil lifespan can be influenced by the specific viability or apoptosis assay employed. We tested the viability of neutrophils by Calcein-AM staining at different time points in vitro (**see Fig6 for reviewers**). The

viability assay with Calcein-AM relies on the ability of this dye to enter live cells where it is cleaved by intracellular esterases, converting it into the fluorescent dye calcein, which is retained within viable cells. Only viable cells with intact plasma membranes retain fluorescence, making this a true end-point assay for cell viability. While viable cells exhibit strong green fluorescence, early apoptotic cells can still be stained light green, and thus were considered live cells during our previous counting process. We realized that this counting method might have included early apoptotic cells with light green fluorescence, which may have biased our results.

Since green fluorescence intensity is proportional to the number of viable cells, we re-analyzed the images by quantifying the fluorescence intensity of the cultured neutrophils. The mean fluorescence intensity of the cells was calculated and compared to the fluorescence intensity of the positive control, which reflected coherent changes in viability (**Revised Fig3 in the manuscript**).

The viability of neutrophils was determined and quantified based on Calcein-AM staining at 4h, 8h, 12h, 18h and 24h after tPA treatment (**Fig.6 for reviewers**). The results indicated that neutrophil viability was around $77.02 \pm 2.14\%$ at 12h, suggesting that they can be used for general in vitro experimental research (**Fig.6D, G for reviewers**). The viability rate of tPA-treated neutrophils continued to decrease beyond 12 h of culture, whereas the protective effect of HRG toward neutrophil viability could be clearly observed at 18 h ($58.07 \pm 1.51\%$ vs 79.38 ± 0.64 for tPA vs tPA+HRG) (**Fig.6E for reviewers**). Therefore, the data for the viability assay at 18 h of incubation was presented in the manuscript.

Figure 6 for reviewers (also shown as Appendix Figure S3 in revised manuscript). Neutrophils viability determined at different time points after tPA and/or HRG treatment in vitro. (A) Neutrophils were labeled with calcein-AM (living cell, green) and Hoechst 33342 (nuclei, blue). tPA and HRG were added to the culture system with LPS (10 $\mu\text{g/ml}$) added as the positive control. After 4 h, 8 h, 12 h, 18 h and 24 h of culture, the viability of neutrophils was observed and calculated under a fluorescent microscope. Typical results from five independent experiments are shown. Scale bar = 20 μm . (B-F). Statistical analysis of the cell viability rate at 4 h, 8 h, 12 h, 18 h and 24 h of culture is shown (n = 5-6). (G) The line graph summarizes the cell viability at different time points (n = 5-6). Data information: Results are shown as mean \pm SEM. Two tailed Kruskal-Wallis H test is

used in (B-F).

As for the migration assay, since the viability of neutrophils was still around $77.02 \pm 2.14\%$ at 12 h of culture, we monitored the migration up to 8 h in the transwell assay after a 2 h tPA and/or HRG pre-treatment. Even though the percentage of migrated neutrophils continued increase at 4-8h, the migration rate gradually decreased as shown in **Fig.7E, F for reviewers**. The migration rate was highest within the first 2 hours in the transwell assay, but decreased gradually at 2-4 hours and even more profoundly at 4-8 hours (**Fig.7E, F for reviewers, also shown as Fig.5G, H in the revised manuscript**). The decreased migration rate could be attributed to impaired neutrophils viability during prolonged incubation. However, the effect of HRG on inhibiting neutrophils migration could be reflected by the total number of migrated cells at 8 h, which represents the proportion of migrated cells over the particular time period (**Fig. 7B-D for reviewers**). Therefore, the migration assay with the indicated proportion and rate of cell migration at different time point was shown in the revised manuscript.

Figure 7 for Reviewers (also shown as Figure 5C-H in revised manuscript). HRG inhibits neutrophil migration across the BBB in vitro. (A) Schematic diagram illustrating the experimental design. Transwell membrane inserts were coated with collagen IV and fibronectin, and hCMEC/D3 cells were seeded onto them followed by 4 h of hypoxia-glucose deprivation. Neutrophils pretreated with different factors (tPA: 10 µg/ml, HRG: 0.1 µM, HRG: 0.5 µM, and HRG: 1 µM) for 2 h were added to the upper chamber. After 2 h (B) (n = 8), 4 h (C) (n = 7-8), and 8 h (D) (n = 7-8) of culture, the number of neutrophils in the lower chamber was counted. Migration percentage = the number of neutrophils in the lower chamber/the total number of neutrophils added to the upper chamber. (E) The line graph summarizes the migrated percentage of neutrophils at 2 h, 4 h and 8 h (n = 6-8). (F) The average migration rate of neutrophils during the 0-2 h, 2-4 h and 4-8 h of the transwell assay. Data information: Data are presented as mean ± SEM; Two tailed Kruskal-Wallis H test is used in (B, D-F).

3. In the literatures, siRNA for HRG has been used to reduce the expression of HRG in the liver and to obtain the acute and efficient lowering of plasma HRG in in vivo experiments. The authors should examine the effects of depletion of endogenous HRG on hemorrhagic transformation in the present model.

Reply: Thank you for the insightful questions raised regarding our study. HRG administration was found protective towards delayed-thrombolysis induced HT and neutrophil infiltration. We used siRNA for depletion of endogenous HRG and investigated the effect on HT. siRNA for HRG was effective in reducing HRG mRNA expression in AML12 cells in vitro (**Fig.3A for reviewers**). Additionally, this treatment reduced the expression of HRG mRNA in liver (**Fig.3B for reviewers**) and efficiently reduced plasma HRG levels in vivo (**Fig.3C for reviewers**). We treated mice with HRG siRNA, induced MCAO, and performed delayed tPA treatment. We found that neutrophil infiltration, HT and neurological deficits were exacerbated in the HRG siRNA treatment group (**Fig.4 for reviewers**).

4. HRG is produced mainly in the liver. Therefore, the authors should provide data on the expression of HRG mRNA after t-PA treatment.

Reply: We appreciate the constructive feedback. HRG mRNA expression levels in liver were determined, which showed increases as early as 1h post-tPA treatment and a slight decrease at 4h post-tPA treatment (**Fig. 1A for reviewers**). Additionally, HRG mRNA could not be detected in immune tissues including the spleen, thymus, lymph node, bone marrow and peripheral blood leucocytes in C57BL/6 mice following tPA treatment (**Fig. 1B for reviewers**).

5. Minor point, page 9, line 17: 3 mm pores? Is this correct?

Reply: We apologize for the oversight and have corrected this issue in the revised manuscript. The corrected description is as following: Transwell membrane inserts (3 μ m pores, 11 mm in diameter; Corning Life Science, Lowell, MA, USA) were coated with collagen IV (50 mg/ml) and fibronectin (30 mg/ml).

Referee #3 :

1. Authors should add in the abstract the meaning of HRG the first time they cite it.

Reply: We apologize for the oversight and have corrected this issue in the revised manuscript.

2. The introduction contains the conclusions of the study. I think the introduction should explain the current state of the art and the reasons leading the authors to investigate but not the results or the conclusions of the study.

Reply: Thanks for the suggestions and we have made the requested changes in the Introduction section.

3. Methodology: in the abstract it is stated that the study was validated in a cohort of 97 patients but then from methodology it results that the three groups considered were quite unbalanced, with 19 healthy volunteers as control, 61 patients receiving tPA and 17 patients without tPA. This could affect the results of the study, so the authors should include similar number of patients for the several groups of the study or perform some statistical analysis considering this bias and include this as limitation.

Reply: We appreciate the valuable feedback and valid concerns. We have recruited additional healthy volunteers as well as patients with acute ischemic stroke who declined tPA treatment. The validation cohorts have included 53 healthy volunteers, 62 patients receiving tPA and 42 patients without tPA. The results and descriptions have been updated in the related content of the manuscript (**Fig 2 in the revised manuscript**).

4. Authors should add tables with the values of HRG in the control and in the 2 patient groups with all the relevant stats

Reply: We now provided a table with the HRG values and relative statistics in the Expanded View Content (**Table EV 2-5**).

5. Kruskal-Wallis test should be used rather than Mann-Whitney U-test as I get you have 3 groups to

compare: control, tPA yes and tPA no.

Reply: Thanks for the suggestions and we apologize for the oversight. We performed Kruskal-Wallis test when we had more than 3 groups to compare. The p values were calculated accordingly and noted on the statistical graphs.

References:

- Castellanos M, Leira R, Serena J, Blanco M, Pedraza S, Castillo J, Davalos A (2004) Plasma cellular-fibronectin concentration predicts hemorrhagic transformation after thrombolytic therapy in acute ischemic stroke. *Stroke* 35: 1671-6
- Castellanos M, Sobrino T, Millan M, Garcia M, Arenillas J, Nombela F, Brea D, Perez de la Ossa N, Serena J, Vivancos J, Castillo J, Davalos A (2007) Serum cellular fibronectin and matrix metalloproteinase-9 as screening biomarkers for the prediction of parenchymal hematoma after thrombolytic therapy in acute ischemic stroke: a multicenter confirmatory study. *Stroke* 38: 1855-9
- Foerch C, Wunderlich MT, Dvorak F, Humpich M, Kahles T, Goertler M, Alvarez-Sabin J, Wallesch CW, Molina CA, Steinmetz H, Sitzer M, Montaner J (2007) Elevated serum S100B levels indicate a higher risk of hemorrhagic transformation after thrombolytic therapy in acute stroke. *Stroke* 38: 2491-5
- Gao S, Wake H, Sakaguchi M, Wang D, Takahashi Y, Teshigawara K, Zhong H, Mori S, Liu K, Takahashi H, Nishibori M (2020) Histidine-Rich Glycoprotein Inhibits High-Mobility Group Box-1-Mediated Pathways in Vascular Endothelial Cells through CLEC-1A. *iScience* 23: 101180
- Gori AM, Giusti B, Piccardi B, Nencini P, Palumbo V, Nesi M, Nucera A, Pracucci G, Tonelli P, Innocenti E, Sereni A, Sticchi E, Toni D, Bovi P, Guidotti M, Tola MR, Consoli D, Micieli G, Tassi R, Orlandi G et al. (2017) Inflammatory and metalloproteinases profiles predict three-month poor outcomes in ischemic stroke treated with thrombolysis. *J Cereb Blood Flow Metab* 37: 3253-3261
- Hulett MD, Parish CR (2000) Murine histidine-rich glycoprotein: cloning, characterization and cellular origin. *Immunology and cell biology* 78: 280-7
- Inzitari D, Giusti B, Nencini P, Gori AM, Nesi M, Palumbo V, Piccardi B, Armillis A, Pracucci G, Bono G, Bovi P, Consoli D, Guidotti M, Nucera A, Massaro F, Micieli G, Orlandi G, Perini F, Tassi R, Tola MR et al. (2013) MMP9 variation after thrombolysis is associated with hemorrhagic transformation of lesion and death. *Stroke* 44: 2901-3
- Karlinski M, Bembenek J, Grabska K, Kobayashi A, Baranowska A, Litwin T, Czlonkowska A (2014) Routine serum C-reactive protein and stroke outcome after intravenous thrombolysis. *Acta Neurol Scand* 130: 305-11
- Koide T, Foster D, Yoshitake S, Davie EW (1986) Amino acid sequence of human histidine-rich glycoprotein derived from the nucleotide sequence of its cDNA. *Biochemistry* 25: 2220-5
- Leung LL, Harpel PC, Nachman RL, Rabellino EM (1983) Histidine-rich glycoprotein is present in human platelets and is released following thrombin stimulation. *Blood* 62: 1016-21
- Montaner J, Molina CA, Monasterio J, Abilleira S, Arenillas JF, Ribo M, Quintana M, Alvarez-Sabin J (2003) Matrix metalloproteinase-9 pretreatment level predicts intracranial hemorrhagic complications after thrombolysis in human stroke. *Circulation* 107: 598-603
- Ning M, Sarracino DA, Buonanno FS, Krastins B, Chou S, McMullin D, Wang X, Lopez M, Lo EH (2010) Proteomic Protease Substrate Profiling of tPA Treatment in Acute Ischemic Stroke Patients: A Step Toward Individualizing Thrombolytic Therapy at the Bedside. *Transl Stroke Res* 1: 268-75
- Nishimura Y, Wake H, Teshigawara K, Wang D, Sakaguchi M, Otsuka F, Nishibori M (2019) Histidine-rich glycoprotein augments natural killer cell function by modulating PD-1 expression via CLEC-1B. *Pharmacology research & perspectives* 7: e00481
- Poon IK, Patel KK, Davis DS, Parish CR, Hulett MD (2011) Histidine-rich glycoprotein: the Swiss Army knife of mammalian plasma. *Blood* 117: 2093-101
- Saito H, Goodnough LT, Boyle JM, Heimburger N (1982) Reduced histidine-rich glycoprotein levels in plasma of patients with advanced liver cirrhosis. Possible implications for enhanced fibrinolysis. *The American journal of medicine* 73: 179-82
- Sia DY, Rylatt DB, Parish CR (1982) Anti-self receptors. V. Properties of a mouse serum factor that blocks

autorosetting receptors on lymphocytes. *Immunology* 45: 207-16

Takahashi Y, Wake H, Sakaguchi M, Yoshii Y, Teshigawara K, Wang D, Nishibori M (2021) Histidine-Rich Glycoprotein Stimulates Human Neutrophil Phagocytosis and Prolongs Survival through CLEC1A. *J Immunol* 206: 737-750

5th Jul 2024

Dear Prof. Shi,

Thank you for the submission of your revised manuscript to EMBO Molecular Medicine. We have now received the enclosed reports from the referees that were asked to re-assess it. As you will see the reviewers are now globally supportive and I am pleased to inform you that we will be able to accept your manuscript pending the following final amendments:

- 1) Please check the "Author Checklist" carefully and complete all relevant questions. Currently a response for whether the data are described by technical or biological replicates is missing (row 88, column D).
- 2) Data availability: Please remove the reviewer access codes from the Data availability statement and ensure that the datasets are now freely accessible. You may remove the statement that the data were deposited according to ethical guidelines and with informed consent as it is in the appropriate Materials and Methods section (although it is also fine to leave it as is).
- 3) Please rename "Conflict of Interest" to "Disclosure and competing interests statement". We updated our journal's competing interests policy in January 2022 and request authors to consider both actual and perceived competing interests. Please review the policy <https://www.embopress.org/competing-interests> and update your competing interests if necessary.
- 4) References: Please correct the reference citation in the reference list such that when there are more than 10 authors on a paper, only the first 10 should be listed, followed by "et al.". Please check "Author Guidelines" for more information. <https://www.embopress.org/page/journal/17574684/authorguide#referencesformat>
- 5) In the Materials and Methods, please ensure that catalog numbers for the antibodies used for the in vitro NETosis assay are reported. Currently these are missing for H3Cit, MPO, and the two secondary antibodies.
- 6) All Materials and Methods need to be described in the main text using our 'Structured Methods' format, which is required for all research articles. According to this format, the Methods section includes a Reagents and Tools Table (listing key reagents, experimental models, software and relevant equipment and including their sources and relevant identifiers) followed by a Methods and Protocols section describing the methods using a step-by-step protocol format. The aim is to facilitate adoption of the methodologies across labs. More information on how to adhere to this format as well as a downloadable template (.docx) for the Reagents and Tools Table can be found in our author guidelines: <https://www.embopress.org/page/journal/17574684/authorguide#structuredmethods>
An example of a Method paper with Structured Methods can be found here: <https://www.embopress.org/doi/10.15252/msb.20178071>.
- 7) Please place individual sections of the manuscript in the following order: Title page - Abstract & Keywords - Introduction - Results - Discussion - Materials & Methods - Data Availability - Acknowledgements - Disclosure and Competing Interests Statement - The Paper Explained - For More Information - References - Figure Legends - Expanded View Figure Legends.
- 8) For the figures and figure legends, please take care of the following:
 - Please indicate the statistical test used for data analysis in the legends of figures 7a-b, e-f; EV 4d; EV 5a-b.
 - Please indicate the statistical test used for data analysis in the legends of figures 7a-b, e-f; EV 4d; EV 5a-b.
 - Please note that the scale bar needs to be defined for figure EV 3a.
 - Please make sure that the callouts of the figures are in sequential order. Currently Appendix Figure S1 is called out after all of the other appendix figures.
- 9) Tables: Please upload all EV tables separately.
- 10) Appendix file: Please upload the Appendix as a single PDF (no separate image files are needed) and add a Table of Contents with page numbers for each figure/table.
- 11) Funding: Please double check that all funding sources are entered into the manuscript submission system - currently it is unclear to us whether the Tianjin Key Medical Discipline (Specialty) Construction Project should be listed in our system.
- 12) Synopsis:
 - Synopsis text: Please provide a short standfirst (maximum of 300 characters, including space), limit the bullet points to max. 5 and upload it as a separate .doc file. Please write the bullet points to summarise the key NEW findings. They should be designed to be complementary to the abstract - i.e. not repeat the same text. We encourage inclusion of key acronyms and quantitative information (maximum of 30 words / bullet point). Please use the passive voice.
 - Please check your synopsis text and image before submission with your revised manuscript. Please be aware that in the proof stage minor corrections only are allowed (e.g., typos).
- 13) For more information: This space should be used to list relevant web links for further consultation by our readers. Could you identify some relevant ones and provide such information as well? Some examples are patient associations, relevant databases, OMIM/proteins/genes links, author's websites, etc...
- 14) As part of the EMBO Publications transparent editorial process initiative (see our policy here: https://www.embopress.org/transparent-process#Review_Process), EMBO Molecular Medicine will publish online a Peer Review File (PRF) to accompany accepted manuscripts. This file will be published in conjunction with your paper and will include the anonymous referee reports, your point-by-point response and all pertinent correspondence relating to the manuscript. Let us know whether you agree with the publication of the PRF and as here, if you want to remove or not any figures from it prior to publication. Please note that the Authors checklist will be published at the end of the PRF.
- 15) Please provide a point-by-point letter INCLUDING my comments as well as the reviewer's reports and your detailed

responses (as Word file).

I look forward to reading a new revised version of your manuscript as soon as possible.

Yours sincerely,

Poonam Bheda

Poonam Bheda, PhD
Scientific Editor
EMBO Molecular Medicine

***** Reviewer's comments *****

Referee #1 (Remarks for Author):

Overall, the authors have sufficiently addressed all of my comments. They provided comprehensive responses, added new data and figures, and made necessary revisions to the manuscript to clarify and support their findings.

Referee #2 (Remarks for Author):

The authors have responded to my comments properly and revised the manuscript incorporating the additional data. Especially, the methodological concerns on the viability of neutrophils have been solved.

The authors addressed the minor editorial issues.

29th Jul 2024

Dear Prof. Shi,

Congratulations on an excellent manuscript, I am pleased to inform you that your manuscript has been accepted for publication in the EMBO Molecular Medicine. Thank you for your comprehensive response to referee concerns and for providing detailed source data. It has been a pleasure to work with you to get this to the acceptance stage.

Yours sincerely,

Poonam Bheda, PhD
Scientific Editor
EMBO Molecular Medicine
